# *GeoDynamics*: A Geometric State-Space Neural Network for Understanding Brain Dynamics on Riemannian Manifolds

**Tingting Dan**        **Jiaqi Ding**        **Guorong Wu**$^{*}$

Departments of Psychiatry and Computer Science
University of North Carolina at Chapel Hill
Chapel Hill, NC 27599
{Tingting_Dan,grwu}@med.unc.edu;jiaqid@cs.unc.edu

## Abstract

State-space models (SSMs) have become a cornerstone for unraveling brain dynamics, capturing how latent neural states evolve over time and give rise to observed signals. By combining deep learning's flexibility with SSMs' principled dynamical structure, recent studies have achieved powerful fits to functional neuroimaging data. However, most approaches still view the brain as a set of loosely connected regions or impose oversimplified network priors, falling short of a truly holistic, self-organized dynamical system perspective. Brain functional connectivity (FC) at each time point naturally forms a symmetric positive definite (SPD) matrix, which lives on a curved Riemannian manifold rather than in Euclidean space. Capturing the trajectories of these SPD matrices is key to understanding how coordinated networks support cognition and behavior. To this end, we introduce *GeoDynamics*, a geometric state space neural network that tracks latent brain state trajectories directly on the high-dimensional SPD manifold. *GeoDynamics* embeds each connectivity matrix into a manifold-aware recurrent framework, learning smooth, geometry-respecting transitions that reveal task-driven state changes and early markers of Alzheimer's, Parkinson's, and autism. Beyond neuroscience, we validate *GeoDynamics* on human action recognition benchmarks (UTKinect, Florence, HDM05), demonstrating its scalability and robustness in modeling complex spatiotemporal dynamics across diverse domains.

## 1 Introduction

The human brain is a complex, dynamic system with distinct structural regions specialized for specific functions, which are locally segregated yet interconnected to process diverse information [36]. Over recent decades, understanding the brain's functional mechanisms has been a central focus in neuroscience. Functional magnetic resonance imaging (fMRI), a widely used non-invasive technique, measures blood oxygen level-dependent (BOLD) signals over time, which are linked to neural activity. While initial research emphasized BOLD signals, the focus has shifted toward functional connectivity (FC), which captures co-activations across brain regions [3]. Studies using Pearson's correlation to quantify FC have shown that functional brain networks are not static but exhibit dynamic topology changes, even in task-free states [4]. These dynamic changes have been associated with brain disorders, offering insights into underlying neurobiological processes [8].

Efforts to model brain dynamics have focused on two main approaches: (1) analyzing temporal fluctuations in BOLD signals and (2) capturing topology changes in evolving FC matrices. While

---

$^{*}$Corresponding author.

39th Conference on Neural Information Processing Systems (NeurIPS 2025).

BOLD signals track neural activity, they often struggle with noise and intrinsic fluctuations. Neural mass models, for instance, describe brain dynamics using non-linear equations but often ignore spatial dependencies [63]. FC matrices, on the other hand, reveal network-level interactions and, when extended to dynamic FC (dFC), track connectivity evolution over time using methods like sliding windows [42]. Recent advances, such as a geometric-attention neural network proposed in [15], relate FC topology changes to brain activity. However, sliding window techniques remain sensitive to window size, which can hinder the detection of subtle brain state changes.

The widespread success of recurrent neural networks (RNNs, Fig. 1 (a)) [59], including long short-term memory (LSTM) [31] and gated recurrent units (GRU) [14], in sequential modeling tasks such as natural language processing (NLP), has inspired numerous efforts to apply these architectures for characterizing brain dynamics [47; 48]. Recently, state space models (SSMs) (as shown in Fig.1 (b, black solid box)) [27; 26] have emerged as a powerful tool for capturing a system's behavior using hidden variables, or "states", marked as $s_t$ (i.e., $s(t)$), which effectively model temporal dependencies in sequential data with well-established theoretical properties. These models have gained significant attraction in fields like computer vision (CV) and natural language processing (NLP) due to their ability to represent complex temporal patterns. A more inclusive literature survey can be found in the Appendix A.1.

**Relevant work of SSM on brain functional studies.** While SSMs have achieved success in CV and NLP applications, their use in brain functional studies has primarily focused on event-related fMRI data [22; 35], limiting their applicability to resting-state fMRI (rs-fMRI). To address this, [64] combined an auto-encoder for learning BOLD signal relationships with a hidden Markov model (HMM) for state transitions, but the separate training of these components reduced efficiency and neglected brain network spatial structures. Similarly, [66] proposed a linear SSM using variational Bayesian methods to infer effective connectivities from EEG and fMRI data. However, this approach struggles to capture the complex dynamics between evolving functional connectivity and underlying cognitive or behavioral outcomes.

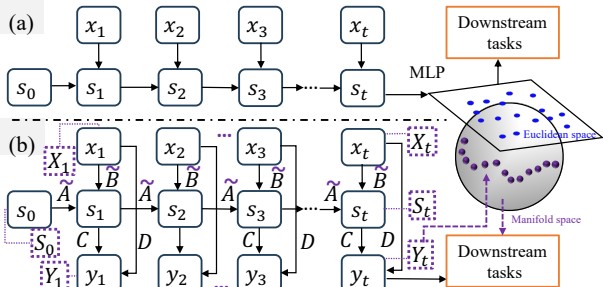

Figure 1: The architecture of RNNs (a) typically relies on a multi-layer perceptron (MLP) to project the system state space into the output space, where various downstream tasks are then performed. These models operate entirely within Euclidean space. In contrast, vanilla SSMs (b, black solid box) incorporate two ordinary differential equations (ODEs), the state equation (upper) and observation equation (lower), which can directly perform downstream tasks through the inferred observed output, also within *Euclidean space*, focusing primarily on temporal information. Our geometric deep model of SSM (b, purple dashed box) extends this approach by capturing both temporal and spatial information, operating on a *manifold space*.

**Our work.** The dynamic nature of complex system cannot be understood by thinking of the system as comprised of independent elements. Rather, an approach is needed to utilize knowledge about the complex interactions within a system to understand the behavior of the system overall. In light of this, modeling the fluctuation of functional connectivities on the Riemannian manifold provides a holistic view of understanding how brain function emerges in cognition and behavior. In this paper, we integrate the power of geometric deep learning on Riemannian manifold and the mathematical insight of SSM to uncover the interplay between evolving brain states and observed neural activities. *First*, our method is *structural* in that we propose to learn intrinsic FC feature representations on the Riemannian manifold of symmetric positive–definite (SPD) matrices [17], which allows us to take the whole-brain wiring patterns into account by considering each FC matrix as a manifold instance. *Second*, our method is *behavioral* in that we leverage the SSM to model temporal dynamics. As shown in Fig. 1 (b), SSMs operate through two core ODEs, the state equation and the observation equation, which describe the relationship between the input $x(t)$ (short for $x_t$) of the dynamic system and the system output $y(t)$ (short for $y_t$) at a given time $t$, mediated by a latent state $s(t)$ (short for $s_t$). Taken together, **our contributions** have threefold. (1) We present a novel geometric deep model by integrating state space model and manifold learning. By incorporating Riemannian geometry, our deep model provides an in-depth insight into system dynamics and state transitions, enhancing the model's ability to capture both temporal and spatial complexities in a data-driven manner. (2) We replace

the Euclidean algebra of conventional SSMs with Riemannian geometric algebra (accompanied by theoretical analysis) to effectively capture the spatio-temporal information, which allows us to better handle irregular data structures and harness the geometric properties of SPD matrices. (3) We have significantly improved the computational efficiency compared to manifold-based deep models by using modern machine techniques such as geometric deep model (Sec. 3.1) and SPD-preserving attention mechanism (Sec. 3.2).

We apply our proposed method (*GeoDynamics*) to two types of system dynamics: brain dynamics and action recognition [6; 28]. While brain dynamics is our primary focus, action recognition serves as a validation task to assess the method's generalization performance across different domains. In the application of understanding brain dynamics, upon which we refer to as *GeoDynamics*, we have evaluated model performance on the large-scale human brain connectome (HBC) databases, including one Human Connectome Project [79] and four disease-related resting-state fMRI data: (1) Alzheimer's Disease Neuroimaging Initiative (ADNI) [52], (2) Open Access Series of Imaging Studies (OASIS) [46], (3) Parkinson's Progression Markers Initiative (PPMI) [50], and (4) the Autism Brain Imaging Data Exchange (ABIDE). For action recognition, we use three classic human action recognition (HAR) datasets including the Florence 3D Actions dataset [61], the HDM05 database [53], and the UTKinect-Action3D (UTK) dataset [75]. Our *GeoDynamics* has achieved significant results across both brain dynamics and action recognition tasks, demonstrating its effectiveness and practicality. These applications on both neuroscience and computer vision highlight the scalability and robustness of *GeoDynamics* in understanding complex spatio-temporal dynamics across diverse systems.

## 2   Preliminary

### 2.1   Classical State Space Model

The continuous-time state space model is defined as

$$s'(t) = A\, s(t) + B\, x(t) \quad \text{and} \quad y(t) = C\, s(t) + D\, x(t), \tag{1}$$

where $s(t) \in \mathbb{R}^N$ is the system state, $x(t) \in \mathbb{R}$ is the control input, and $y(t) \in \mathbb{R}^M$ the output (we follow the single-input/single-output convention unless otherwise stated). The matrices $A \in \mathbb{R}^{N \times N}$, $B \in \mathbb{R}^{N \times 1}$, $C \in \mathbb{R}^{M \times N}$, and $D \in \mathbb{R}^{M \times 1}$ (often $D = 0$) encode transition and observation, as sketched in Fig. 1(b). While Eq. (1) assumes Euclidean variables, many neuroscientific signals, notably functional connectivity matrices, are naturally manifold-valued (SPD), motivating a geometry-aware extension.

### 2.2   Riemannian Geometry on the SPD Manifold

Let $\mathcal{M} = \mathrm{SPD}(N)$ denote the manifold of $N \times N$ SPD matrices. We write $X, Y \in \mathcal{M}$ and use the Stein metric [11] for computational efficiency:

$$d(X, Y) = \sqrt{\log \det\left(\frac{X + Y}{2}\right) - \tfrac{1}{2} \log \det(XY)}. \tag{2}$$

This distance avoids repeated eigendecompositions while preserving key geometric structure.

**Group action and isometric "translation".**   Let $\mathbb{G} = \mathrm{GL}(N)$ be the general linear group acting on $\mathcal{M}$ via $g.X := gXg^\top$. Under the Stein metric, such actions are isometric, restricting to the orthogonal subgroup $\mathbb{O}(N) \subset \mathbb{G}$ preserves distances, $d(g.X, g.Y) = d(X, Y)$ for all $X, Y \in \mathcal{M}$ (proof in Appendix A.2). We call this an isometric *translation* on $\mathcal{M}$ and denote it by

$$\mathcal{T}_X(g) := gXg^\top, \qquad g \in \mathbb{O}(N). \tag{3}$$

**Weighted Fréchet mean (wFM).**   Given $\{X_n\}_{n=1}^N \subset \mathcal{M}$ and nonnegative weights $\{w_n\}_{n=1}^N$ with $\sum_n w_n = 1$, the wFM is

$$\mathcal{F}(\{X_n\}, \{w_n\}) = \arg\min_{F \in \mathcal{M}} \sum_{n=1}^N w_n\, d^2(X_n, F). \tag{4}$$

Assuming $\{X_n\}$ lies in a suitable geodesic ball, existence/uniqueness holds (see Ref. [1]).

**SPD convolution on $\mathcal{M}$.** For matrix-valued features, the SPD convolution at layer $k$ is

$$X_{i,j}^{(k)} = \sum_{u=0}^{\theta-1} \sum_{v=0}^{\theta-1} H_{u,v} \, X_{i+u,\,j+v}^{(k-1)}, \tag{5}$$

with kernel $H \in \mathbb{R}^{\theta \times \theta}$. To preserve SPD structure, we parameterize

$$H = Z^\top Z + \epsilon I, \qquad \epsilon \to 0^+, \tag{6}$$

so the learned parameter is $Z$ while $H$ is guaranteed SPD, consequently $X^{(k)}$ remains SPD (proof in Appendix A.3).

## 3 Method

**Geometric formulation of state space dynamics.** We extend the classical state space formulation to the Riemannian manifold of SPD matrices. In this setting, the *system input $X(t)$*, *system state $S(t)$*, and *system output $Y(t)$* are no longer vectors in $\mathbb{R}^N$, but elements of the SPD manifold $\mathcal{M} = \mathrm{SPD}(N)$, equipped with the Stein metric. This design enables the system dynamics to evolve in a geometrically consistent manner that respects the intrinsic structure of functional connectivity representations. Formally, at time $t$, we denote: $X(t) \in \mathcal{M}$, $S(t) \in \mathcal{M}$, $Y(t) \in \mathcal{M}$. The temporal evolution of the system is driven by a set of learnable operators defined on $\mathcal{M}$, replacing the Euclidean linear operators of conventional SSMs.

### 3.1 State Update and Observation on the SPD Manifold

Let $\tau$ denote the length of the temporal receptive field (i.e., sliding window). We model the evolution of $S^{(k)}$ and $Y^{(k)}$ at discrete time step $k \in \{1, K\}$ through two components: (1) *Temporal aggregation* of past states and inputs using the weighted Fréchet mean (wFM) to ensure intrinsic averaging along geodesics rather than Euclidean linear combinations. (2) *Group action translation* on the manifold to represent state transitions and input effects in an isometry-preserving manner.

The state update and observation equations are defined as:

$$S^{(k)} = \mathcal{T}\Big(\mathcal{F}\big(\{S_j\}_{j=k-\tau}^{k-1}, \{A_j\}_{j=k-\tau}^{k-1}\big), \ \mathcal{F}\big(\{X_j\}_{j=k-\tau}^{k}, \{B_j\}_{j=k-\tau}^{k}\big)\Big),$$

$$Y^{(k)} = \mathcal{T}\Big(\mathcal{F}\big(\{S_j\}_{j=k-\tau}^{k}, \{C_j\}_{j=k-\tau}^{k}\big), \ \mathcal{F}\big(\{X_j\}_{j=k-\tau}^{k}, \{D_j\}_{j=k-\tau}^{k}\big)\Big), \tag{7}$$

where $\mathcal{F}$ denotes the wFM operator on $\mathcal{M}$ (Eq. 4), and $\mathcal{T}$ denotes the translation on the manifold induced by orthogonal group actions (Eq. 3). The learnable parameters $\{A_j, B_j, C_j, D_j\}$ control how the history of states and inputs contributes to the current state.

**Weighted Fréchet mean aggregation.** Given a set of SPD matrices $\{Z_j\}_{j=k-\tau}^{k}$ and nonnegative weights $\{w_j\}_{j=k-\tau}^{k}$ with $\sum_j w_j = 1$, the weighted Fréchet mean is defined as:

$$\mathcal{F}\big(\{Z_j\}, \{w_j\}\big) = \arg \min_{F \in \mathcal{M}} \sum_{j=k-\tau}^{k} w_j \, d^2(Z_j, F), \tag{8}$$

where $d(\cdot, \cdot)$ is the Stein geodesic distance on $\mathcal{M}$. Unlike Euclidean averaging, this operation produces an intrinsic barycenter on the manifold, ensuring that the aggregated representation remains SPD.

**Group action translation.** To propagate states forward in time, we introduce a translation operator $\mathcal{T}$:

$$\mathcal{T}(U, V) = g(V) \, U \, g(V)^\top, \tag{9}$$

where $g(V)$ is an orthogonal transformation inferred from $V$. Since $g(V) \in \mathbb{O}(N)$, this operation is an isometry under the Stein metric and guarantees that the output remains SPD. The translation acts as a multiplicative update on the manifold, analogous to additive transitions in Euclidean SSM.

**Discretized dynamics and control.** We parameterize the temporal dynamics by discretizing the continuous-time operator using a matrix exponential scheme. Specifically, for each step $\Delta$:

$$\widetilde{A} = \exp(\Delta A), \qquad \widetilde{B} = \Delta A^{-1}\big(\exp(\Delta A) - I\big)\Delta B. \tag{10}$$

This formulation ensures stable temporal integration while preserving geometric structure, allowing smooth evolution of the system's hidden states on $\mathcal{M}$. The role of $\widetilde{A}$ is to propagate the previous hidden state, while $\widetilde{B}$ injects control-dependent information from the input. Their influence is integrated within the wFM in Eq. (7) to produce a manifold-consistent transition.

**Task readout via logarithmic mapping.** The output $Y^{(K)}$ is an SPD matrix, which cannot be directly passed to a standard classifier. We therefore map it to the tangent space at the identity using the logarithmic map:

$$\hat{y}^{(K)} = \log\big(Y^{(K)}\big) = \Phi \log(\Lambda)\Phi^{\top}, \tag{11}$$

where $Y^{(K)} = \Phi\Lambda\Phi^{\top}$ is the eigendecomposition. This yields a symmetric matrix in Euclidean space that preserves local manifold geometry.

For classification task and brain state decoding, we employ a softmax layer:

$$P(Q = q \mid \hat{y}^{(K)}) = \frac{\exp(w_q^{\top}\mathrm{vec}(\hat{y}^{(K)}))}{\sum_{q'=1}^{Q}\exp(w_{q'}^{\top}\mathrm{vec}(\hat{y}^{(K)}))}, \tag{12}$$

where $\mathrm{vec}(\cdot)$ denotes vectorization of the symmetric matrix and $Q$ is the number of task classes or clinical outcomes. Model training uses the cross-entropy loss:

$$\mathcal{L} = -\frac{1}{E}\sum_{i=1}^{E}\sum_{q=1}^{Q} o_{iq} \log P(Q = q \mid \hat{y}_i^{(K)}) \tag{13}$$

where $o_{iq}$ is the ground truth label. $E$ denotes the number of subjects/samples.

**Global convolutional reformulation.** Although the above formulation describes the step-wise state evolution, it can be equivalently expressed as a convolution operation over time [25]. Expanding the recurrence gives:

$$\mathcal{K} = \big(CB + D, CAB, CA^2B, \ldots, CA^kB, \ldots\big), \qquad y = x * \mathcal{K}. \tag{14}$$

To ensure SPD structure, we constrain each kernel $\mathcal{K}^{r,l}$ to be SPD by parameterizing

$$\hat{\mathcal{K}} = \mathcal{K}^{\top}\mathcal{K} + \epsilon I, \tag{15}$$

where $\epsilon > 0$ is a stability constant. The nonlinearity is introduced via the exponential map $\exp(\cdot)$ on the Riemannian algebra, which guarantees that the output remains SPD after each convolutional block [15]. Frobenius normalization is applied to control eigenvalue growth and maintain numerical stability during training.

## 3.2 SPD-Preserving Attention

To enhance the model's ability to identify informative disease-related spatio-temporal patterns, we introduce a *SPD-preserving attention* (SPA) module embedded within the manifold convolutional backbone. Unlike conventional attention mechanisms that are defined in Euclidean space, SPA operates directly on the SPD manifold and is designed to preserve the positive-definite structure of the signal throughout the attention process. Given the convolutional response $X * \mathcal{K}$ on $\mathcal{M}$, we define the attention weights as

$$\delta(\cdot) = \frac{\exp\big([X * \mathcal{K}]\big)}{\max\big(\exp([X * \mathcal{K}])\big)}, \qquad [X] := \mathrm{diag}(\rho, X), \tag{16}$$

where $\mathrm{diag}(\rho, X)$ pads the borders of the SPD matrix with zeros of size $\rho - 1$ and introduces a small positive diagonal component to ensure strict positive definiteness (see Appendix A.5). The exponential operation $\exp(\cdot)$ plays a role analogous to the sigmoid function, smoothly mapping the weighted response into the interval $[0, 1]$ while retaining manifold consistency. The resulting $\delta(\cdot)$

acts as a *soft mask* that adaptively modulates local connectivity patterns. The attention-weighted representation is then computed as

$$\widetilde{X} = \delta(\cdot) \odot (X * \mathcal{K}), \tag{17}$$

where $\odot$ denotes elementwise multiplication. Because $\delta(\cdot)$ is strictly positive and bounded, and the base convolution is SPD-constrained, $\widetilde{X}$ remains SPD after modulation. This preserves both the local geometry (SPD structure at each time point) and global temporal consistency.

**Disease-relevant pattern localization.** Many neurological and psychiatric disorders, including neurodegenerative diseases such as Alzheimer's, are characterized by *localized but propagating disruptions in functional connectivity*. These disruptions often manifest as: (1) spatially specific *abnormal hubs* or subnetworks (e.g., default mode, limbic regions), (2) temporal shifts in network activation or coupling, (3) altered covariance structure in FC trajectories reflecting pathological propagation. In conventional SSM or GNN models, such disease-related signals may be smoothed out by global averaging or fixed-weight message passing. In contrast, the SPA mechanism learns *spatially and temporally adaptive weighting functions* directly on the SPD manifold, enabling: *Enhanced sensitivity to local network deviations*: the multiplicative modulation amplifies abnormal signal components (e.g., hyper- or hypo-connectivity in key subnetworks) while attenuating background fluctuations. *Preservation of intrinsic geometry*: since both the attention and the convolution are SPD-preserving, pathological patterns expressed as topological changes (e.g., altered covariance structure) remain valid on the manifold, avoiding distortions induced by Euclidean projections. *Temporal adaptivity*: $\delta(\cdot)$ is computed at each time step, enabling the network to detect evolving abnormal FC trajectories associated with disease progression. Importantly, the learned attention weights $\delta(\cdot)$ are anatomically and temporally interpretable, i.e., high attention scores highlight regions or subnetworks where deviations from healthy connectivity trajectories are most pronounced. This makes SPA not only an effective inductive bias for improving predictive performance, but also a tool for identifying candidate disease biomarkers.

### 3.3 Summary of the *GeoDynamics* Framework

The proposed *GeoDynamics* integrates: *(1) Geometric state evolution:* Input, system states, and outputs are represented as SPD matrices evolving on $\mathcal{M}$, ensuring geometric consistency. *(2) Manifold-aware operators:* Temporal aggregation via weighted Fréchet means and state transitions via orthogonal group actions guarantee that system trajectories remain on the manifold. *(3) Stable discretization:* Matrix exponential schemes provide stable integration and controllable temporal dynamics. *(4) Efficient convolutional formulation:* A global kernel expansion enables parallel training and scalable implementation. *(5) Geometric attention:* SPA enhances interpretability and sensitivity to task-relevant regions.

This formulation provides a principled and computationally efficient way to model non-Euclidean temporal dynamics, making it particularly well suited for functional connectivity sequences and other manifold-valued time series.

## 4 Experiments

### 4.1 Experimental Setup

**Dataset.** We apply our method to two types of datasets including *human brain connectome* (HBC) and *human action recognition* (HAR), more detailed data information is shown in Table 4 of Appendix A.6. *For HBC dataset.* We select one dataset of healthy young adults and four disease-related human brain datasets for evaluation: the HCP Working Memory (HCP-WM) [79], ADNI [52], OASIS [46], PPMI [50], and ABIDE [20]. We selected a total of 1,081 subjects from the HCP-WM dataset. The working memory task included eight task events. Brain activity was parcellated into 360 regions based on the multi-modal parcellation from [24]. For the OASIS (924 subjects) and ABIDE (1,025 subjects) datasets, which are binary-class datasets, one class represents a disease group and the other represents healthy controls. In the ADNI dataset, subjects are categorized based on clinical outcomes into four distinct cognitive status groups. The PPMI dataset also consists of four classes. We employ Automated Anatomical Labeling (AAL) atlas [67] (116 brain regions) on ADNI, PPMI, ABIDE datasets, while Destrieux atlas [18] (160 brain regions) are used in OASIS to verify the scalability

of the models. *For HAR dataset.* We validate the performance of the proposed *GeoDynamics* on three widely-used HAR benchmarks: the Florence 3D Actions dataset [61], the HDM05 database [53], and the UTKinect-Action3D (UTK) dataset [75]. The Florence 3D Actions dataset consists of 9 activities performed by 10 subjects, with each activity repeated 2 to 3 times, resulting in a total of 215 samples. The actions are captured by the motion of 15 skeletal joints. For the HDM05 dataset, we follow the protocol from [71], focusing on 14 action classes. This dataset contains 686 samples, each represented by 31 skeletal joints. Lastly, the UTKinect-Action3D dataset comprises 10 action classes. Each action was performed twice by 10 subjects, yielding a total of 199 samples.

**SPD matrices construction**. *For HBC dataset.* Each fMRI scan has been processed into $N$ mean time courses of BOLD signals, each with $T$ time points (where $N$ represents the number of brain parcellations), we employ a sliding window technique to capture functional brain dynamics. Specifically, we construct a $N \times N$ correlation matrix at each time point $t$ ($t = 1, \ldots, T$) based on the BOLD signal within the sliding window, centered at time $t$. This results in a sequence of FC matrices encoding the functional dynamics for each scan, represented as $\mathcal{X} = \{X(t) \mid t = 1, \ldots, T\} \in \mathbb{R}^{T \times N \times N}$, in Fig. 2 (a). *For HAR dataset.* HAR datasets exhibit variability due to differences in action duration, complexity, the number of action classes, and the technology used for data capture. Therefore, we first apply a preprocessing step following [56] to obtain the SPD matrices. This step involves fixing the root joint at the hip center (red dashed circle in Fig. 2 (b)) and calculating the relative 3D positional differences for all other $N - 1$ joints. For each timestamp $t = 1, \ldots, T$, we obtain a $3 \times (N - 1)$-dimensional column vector representing the relative displacements of the joints. Then, we compute covariance matrices using the method proposed in [56] to yield the SPD matrices. After that, we apply a sliding window technique to capture the dynamics over time, resulting in a sequence of SPD matrices $\mathcal{X} = \{X(t) \mid t = 1, \ldots, T\} \in \mathbb{R}^{T \times (3(N-1)) \times (3(N-1))}$, as illustrated in Fig. 2 (b).

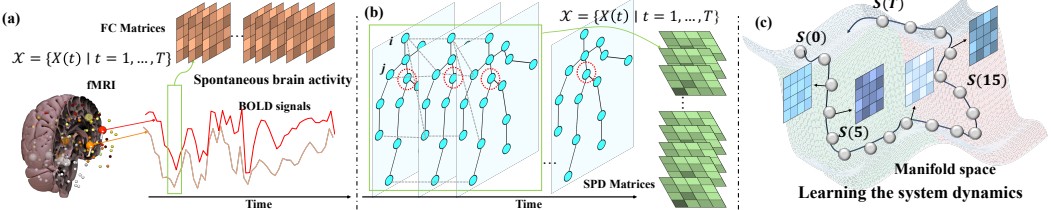

Figure 2: The construction of SPD matrices for HBC (a) and HAR (b) datasets. Learning the system dynamics on manifold space as illustrated in (c).

**Comparison methods and evaluation metrics.** *For HAR dataset.* There are some popular methods for HAR, such as multi-part bag-of-pose (MBP) [61], Lie group [69], shape analysis on manifold (SAM) [19], elastic function coding (EFC) [2], multi-instance multitask learning (MML) [78], Tensor Representation (TR) [45], LieNet[34], SPGK [72], ST-NBNN [73], GR-GCN [23] and DMT-Net and F-DMT-Net [80]. We also include Bi-long short-term memory (Bi-LSTM) [5] and part-aware LSTM (P-LSTM) [62]. *For HBC dataset.* We stratify the comparison methods for HBC into two groups: spatial and sequential models. *Spatial models* focus on capturing brain dynamics. Traditional GNNs like GCN [44] and GIN [77] are included for their ability to handle structured data. Subgraph-based GNNs like Moment-GNN [41] focus on identifying local patterns, while expressive GNNs like GSN [7] and GNN-AK [81] enhance subgraph encoding for better expressivity. SPDNet [16], a manifold-based model, is chosen for managing high-dimensional data. Plus, an MLP serves as a simple, generic baseline. *Sequential models* target temporal dynamics in BOLD signals. 1D-CNN captures temporal patterns, while RNN [59] and LSTM [31] handle sequential dependencies. MLP-Mixer [65] integrates both temporal and spatial information, and Transformer (TF) [68] captures global dependencies through attention. Mamba [25], vanilla SSM, is included for its ability to model system dynamics over time. Two dynamic-FC methods, STAGIN [43], NeuroGraph [60]. Three brain network analysis methods BrainGNN [49], BNT [40], and ContrastPool [76]. More details are shown in Appendix A.7.

**Evaluation metrics.** For Florence and UTKinect datasets, we adopt the standard leave-one-actor-out validation protocol as outlined in [23]. This method generates $Q$ classification accuracy values, which are averaged to produce the final accuracy score. For HDM05 dataset, we follow the experimental setup from [32], conducting 10 random evaluations. In each evaluation, half of the samples from each class are randomly selected for training, with the remaining half used for testing. In all HBC

experiments, we utilize a 10-fold cross-validation scheme, reporting accuracy (Acc), precision (Pre), and F1 score to provide a thorough evaluation of model performance across various datasets.

## 4.2 Results on Human Brain Connectivity Datasets

We investigate brain dynamics across both healthy and disease-related cohorts using task-based and resting-state fMRI data.

**Task-based fMRI.** We first evaluate the task-based working-memory dataset (HCP-WM) using sixteen representative methods. The results in Fig. 3 (green colors) show that sequential models achieve markedly higher accuracy than spatial models (pair-wise $t$-test, $p < 10^{-4}$), with performance gains up to 30%. Our proposed *GeoDynamics* achieves the best overall performance.

*Interpretation.* From a machine learning perspective, sequential models may better capture the temporal dependencies in BOLD signals, which are inherently dynamic during task execution. In contrast, spatial models rely on static functional connectivity patterns that are less sensitive to rapid task-induced variations. Biologically, task-based fMRI paradigms elicit specific neural responses related to cognitive processes such as attention and memory, leading to pronounced fluctuations in brain activity that align naturally with sequential modeling.

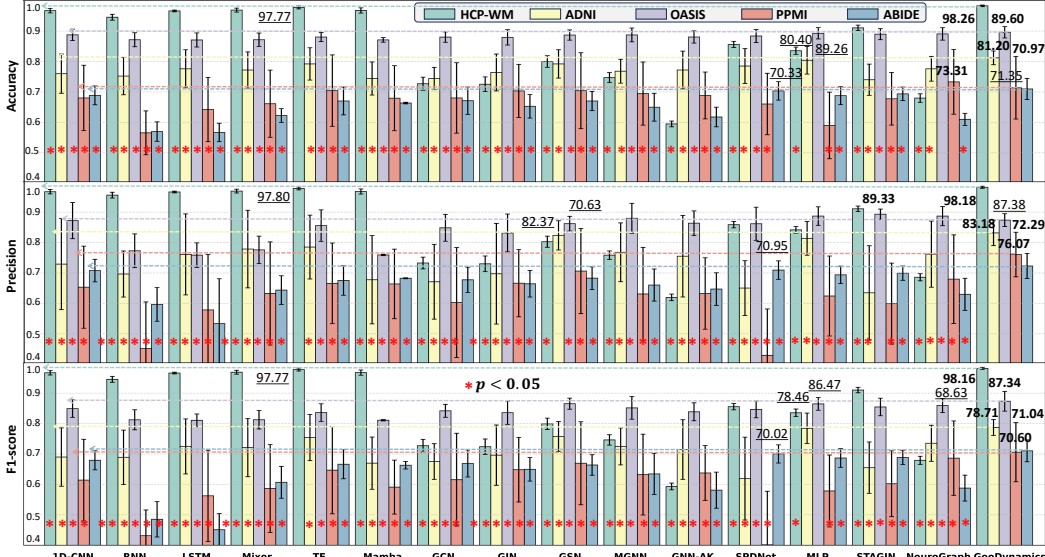

Figure 3: Evaluation performance for different methods across HBC datasets. The best performance is highlighted in bold, while the second-best is underlined.

**Neurodegenerative diseases.** Next, we focus on early diagnosis of neurodegenerative diseases (ND), including Alzheimer's Disease (AD) (yellow and purple colors) and Parkinson's Disease (PD) (pink colors), using resting-state fMRI. We assess the classification between cognitively normal (CN) and neurodegenerative (ND) groups. Spatial models show slightly better average performance than sequential models ($p = 0.37$), although the difference is not statistically significant. Notably, our *GeoDynamics* substantially outperforms all baselines, achieving a significant improvement over the second-best method ($p < 0.05$).

*Interpretation.* Unlike task-based fMRI, resting-state fMRI measures spontaneous neural activity, reflecting intrinsic connectivity rather than stimulus-driven dynamics. This may explain why spatial models perform competitively in ND classification tasks. From a biological standpoint, neurodegeneration primarily manifests as gradual network disconnection rather than transient dysfunction [55; 12]. Functional disruption often precedes measurable cognitive decline by many years [70]. The results in Fig. 3 thus highlight the clinical promise of deep learning models for detecting early ND-related alterations in large-scale brain networks.

**Neuropsychiatric disorders.** We further analyze Autism Spectrum Disorder (ASD) using the ABIDE dataset. As shown in Fig. 3 (blue color), spatial models slightly outperform sequential ones, with SPDNet consistently ranking second only to our *GeoDynamics*. Both SPDNet and *GeoDynamics* share two key design principles: (1) manifold-based representation learning that preserves the intrinsic geometry of functional connectivity matrices (Fig. 2 (c)), and (2) spatio-temporal modeling that

captures the evolution of connectivity patterns over time. The consistent advantage of these two approaches suggests that robust spatio-temporal representation, grounded in Riemannian geometry, is crucial for reliable diagnosis of neuropsychiatric conditions.

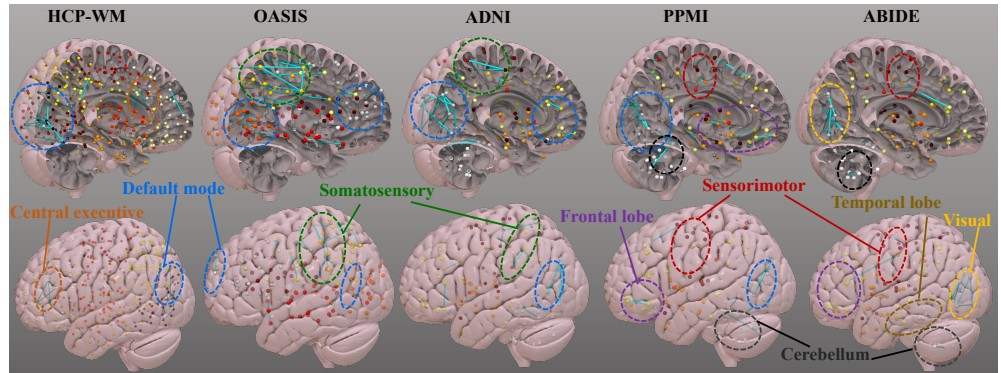

Figure 4: Critical connections from SPD-preserving attention map on HBC datasets.

*Interpretation.* Neuropsychiatric disorders such as Autism and Schizophrenia are characterized by atypical neural connectivity and abnormal variability in BOLD dynamics [58; 51; 38]. These alterations affect both spatial topology and temporal synchronization, suggesting that effective diagnostic models must jointly capture both aspects. In contrast, neurodegenerative diseases primarily disrupt large-scale connectivity due to neuronal loss, making static spatial features more informative. Integrating such disease-specific pathophysiological insights is therefore essential for model design and interpretation.

**Interpretable attention patterns.** Finally, we visualize the attention weights $\delta(\cdot)$ from the SPA module (Sec. 3.2) to identify critical brain regions and connections across datasets. The top-20 strongest connections are mapped onto the brain (Fig. 4). In HCP-WM, the dominant connections reside within the default mode network (DMN, blue) and central executive network (orange), both essential for working-memory tasks. In AD (OASIS and ADNI), the most affected regions include the DMN and somatosensory cortex (green), suggesting that AD may influence sensory processing and bodily awareness in addition to memory loss. In PD, key connections appear in the sensorimotor area (red), frontal lobe (purple), DMN, and cerebellum (black), indicating that PD involves both motor and cognitive-emotional dysfunctions. In Autism, prominent connections emerge in the temporal (brown) and visual cortices (yellow), consistent with known deficits in language, perception, and social interaction.

*Interpretation.* Despite disease heterogeneity, certain shared patterns emerge across neurodegenerative conditions, particularly AD and PD. The proposed attention mechanism not only highlights disease-specific alterations but also uncovers common pathways underlying different disorders. Such interpretability could provide valuable insights into shared pathophysiological mechanisms and guide hypothesis-driven neuroscience research.

### 4.3 Ablation Studies

**Sliding window size (PPMI dataset)**: We examined the effect of varying the sliding window size on model performance. As shown in Table 1, *GeoDynamics* demonstrates relative robustness to window size, achieving optimal performance with moderate values (typically 35–45). This stability likely arises from the SSM module's ability to capture dynamic temporal characteristics, reducing sensitivity to specific window lengths.

Table 1: Ablation studies in terms of sliding window size.

| Window size | 15 | 25 | 35 | 45 | 55 |
|---|---|---|---|---|---|
| Acc | 71.35 ± 10.26 | 70.83 ± 15.74 | 71.69 ± 10.10 | 72.01 ± 8.51 | 71.01 ± 14.23 |
| Pre | 76.07 ± 7.33 | 74.72 ± 10.00 | 73.54 ± 6.50 | 71.73 ± 7.56 | 71.00 ± 7.89 |
| F1 | 70.60 ± 9.73 | 71.71 ± 7.29 | 70.72 ± 3.90 | 68.56 ± 6.74 | 67.67 ± 7.81 |

**Multi-class classification (ADNI dataset)**: We extended the binary classification task to multiple clinical stages. As reported in Table 2, *GeoDynamics* consistently achieves higher accuracy, precision,

and F1-score across all classes compared with both spatial- and sequence-based baselines. This demonstrates robust generalization to more challenging multi-class settings.

Table 2: Multi-class classification results on the ADNI dataset.

| (%) | GCN | GIN | GSN | MGNN | GNN-AK | SPDNet | MLP |
|---|---|---|---|---|---|---|---|
| Acc | 50.00±6.51 | 51.60±5.20 | 52.80±5.31 | 48.80±5.31 | 52.40±6.56 | 52.40±5.20 | 46.40±7.42 |
| Pre | 36.08±14.22 | 39.75±13.50 | 53.23±10.33 | 40.58±10.28 | 46.07±9.48 | 37.01±8.89 | 46.52±12.03 |
| F1 | 38.49±9.73 | 41.76±7.65 | 48.21±7.48 | 38.71±6.11 | 43.20±6.55 | 31.63±8.76 | 43.83±9.13 |
| (%) | 1D-CNN | RNN | LSTM | Mixer | TF | Mamba | *GeoDynamics* |
| Acc | 46.00±5.44 | 45.60±6.25 | 46.00±7.43 | 48.40±4.18 | 52.00±6.93 | 47.20±6.14 | **56.00±3.36** |
| Pre | 36.40±9.72 | 40.95±11.29 | 25.89±11.28 | 48.06±12.84 | 47.63±19.50 | 38.55±13.23 | **60.36±7.67** |
| F1 | 39.21±7.71 | 39.24±7.14 | 31.87±9.93 | 39.40±5.47 | 44.03±11.32 | 37.19±5.53 | **50.83±5.73** |

**Model complexity**: We systematically varied the total number of trainable parameters by adjusting hidden-state dimensionality in Mamba. In the final column in Table 3, we compare *GeoDynamics* against the best-tuned Mamba configurations at the same parameter budget. This demonstrates that our geometry-aware state-space formulation yields measurable gains in predictive performance, even when network capacity is held constant.

Table 3: Comparison between various Mamba configurations and the proposed *GeoDynamics* model on HCP-WM dataset (brain regions $N = 360$). For a fair comparison, the hidden dimension and network depth of Mamba are adjusted to match the parameter scale of *GeoDynamics* (highlighted by underline).

| | **Mamba** | | | | | *GeoDynamics* |
|---|---|---|---|---|---|---|
| **Hidden dim** | 2048 | 1024 | 1024 | 1024 | 512 | $N$ |
| **Network layer** | 5 | 5 | 4 | 2 | 8 | 2 |
| **Para (M)** | 132 | 33.71 | 27.05 | 14.07 | 13.93 | 14.60 |
| **Accuracy** | 97.22±0.63 | 97.06±0.62 | 96.76±0.86 | 95.92±0.50 | 96.17±0.11 | **98.29±0.26** |
| **Precision** | 97.27±0.62 | 97.09±0.60 | 96.80±0.84 | 95.93±0.49 | 96.20±0.10 | **98.18±0.34** |
| **F1-score** | 97.22±0.63 | 97.06±0.62 | 96.76±0.86 | 95.92±0.50 | 96.18±0.11 | **98.16±0.35** |

## 4.4 Model Validation

To evaluate the generality of our framework, we applied it to the HAR dataset using standard joint-coordinate benchmarks (Fig. 5). Despite the differences from neuroimaging data, *GeoDynamics* remains competitive, owing to its unified treatment of spatio-temporal dynamics and manifold geometry. By embedding 3D joint positions on a Riemannian manifold and employing a global convolution kernel, the model captures coordinated movements across distant joints while effectively suppressing noise. This demonstrates the broad applicability of our geometric state-space approach beyond neuroimaging. Furthermore, our method captures higher-order correlations among 3D joint coordinates over time and models spatio-temporal co-occurrences through the tailored convolution kernel, enhancing robustness to noisy joints and improving overall action recognition accuracy.

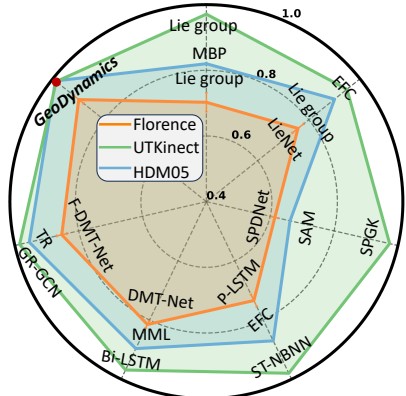

Figure 5: Results on HAR dataset.

More experimental details are provided in Appendix A.7-A.8.

## 5 Conclusion

This work presents a geometric deep model of SSM, *GeoDynamics*, for understanding behavior/cognition through deciphering brain dynamics. In line with theoretical analysis, our method integrates the principles of *geometric deep learning* and efficient feature representation learning on non-Euclidean data, specifically designed for learning on sequential data with inherent topological connections. We have achieved promising experimental results on human connectome data as well as human action recognition, indicating great applicability in real-world data for neuroscience and computer vision.

## Acknowledgement

This work was supported by the National Institutes of Health (AG091653, AG068399, AG084375) and the Foundation of Hope. The views and conclusions contained in this document are those of the authors and should not be interpreted as representing the official policies, either expressed or implied, of the NIH.

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

# A  Appendix

## A.1  Literature Survey

**RNN and its variants on manifold to neuroimaging application.**  Recurrent neural networks (RNNs) have been reformulated as ordinary differential equations (ODEs) with continuous-time system states, as highlighted by LTCNet [30]. These models serve as effective algorithms for modeling time series data and are widely utilized across medical, industrial, and business domains. For instance, [9] has demonstrated its potential for brain state recognition and [29] achieves continuous modeling of dynamic brain signals using ODEs. Furthermore, the survey proposed by [54] provides a comprehensive overview of ODE applications in the field of medical imaging, showcasing their practicality and impact in this domain. Following this, several manifold-based RNN models have emerged. For instance, [10] introduced a statistical recurrent model defined on the manifold of symmetric positive definite (SPD) matrices and evaluated its diagnostic potential for neuroimaging applications. This approach underscores the effectiveness of utilizing manifold-based techniques to enhance the performance of RNNs in complex medical contexts. The RNN model formulated on Riemannian manifolds is robustly supported by mathematical theory, as it utilizes covariance information to dynamically model time-series data [37]. This capability allows it to capture richer and more subtle representations within a higher-dimensional latent space. Such an approach is particularly effective in modeling complex data structures, such as capturing the functional dynamics [15; 33], where the relationships among data points are inherently geometric. By operating within the manifold framework, these models adeptly accommodate the intricacies of underlying data distributions, thereby enhancing both interpretability and predictive performance.

RNNs and their variants, while widely used for sequential modeling tasks, have notable limitations that affect their performance in complex, dynamic systems. One of the key challenges is that RNNs implicitly learn sequential patterns and temporal dependencies, without explicitly modeling the underlying dynamics. This implicit nature makes RNNs harder to interpret, often turning them into "black-box" models where the relationships between input variables and predicted outcomes can be obscured, limiting their utility in scenarios requiring high interpretability. Although advancements like LTCNet [30] have improved the interpretability of RNNs by framing them as an ODE, these models primarily focus on the dynamics of the system states and inputs (as shown in Fig. 1 (a)). However, they failed to consider observation equations (but usually use MLP to fit the observations), which describe the relationship between system states and observed data. This formulation reduces their ability to fully model the observable aspects of a system, resulting in an incomplete picture of the system's dynamics and limiting their explanatory power.

**SSM to neuroimaging application.**  State Space Models (SSMs) explicitly model temporal dynamics through latent variables governed by two key ODEs: the state equation, which captures the evolution of the system state over time, and the observation equation, which relates the latent state to observable data. This structured, ODE-based framework allows SSMs to offer a clearer understanding of how systems evolve and provides a higher level of interpretability compared to RNNs. This makes SSMs particularly valuable in domains requiring an understanding of underlying system dynamics, such as medical diagnostics and time-series forecasting. In contrast to RNNs and their variants (e.g., LSTMs, GRUs), which often operate as "black boxes," SSMs like Kalman Filters [39] have well-established theoretical properties. These properties typically include convergence and stability, providing a solid mathematical foundation that is difficult to guarantee with more complex RNN architectures. RNNs, especially deeper ones, can suffer from issues like vanishing or exploding gradients, which affect training stability and interoperability. SSMs also naturally incorporate probabilistic structures, allowing them to effectively handle noisy or uncertain data. This is particularly advantageous in low Signal-to-Noise Ratio (SNR) datasets, such as fMRI [74] and Electroencephalogram (EEG) data [57], where the ability to account for noise and uncertainty is critical. In light of these performance advantages, only a few manifold-based SSMs have been developed. For instance, [13] explores the modeling of time series observations in state-space forms defined on Stiefel and Grassmann manifolds. This approach utilizes Bayesian methods to estimate state matrices by calculating posterior modes, effectively integrating geometric constraints with probabilistic inference. However, while Bayesian methods excel in handling uncertainty, they often face limitations in scalability, inference speed, and flexibility compared to deep learning models, which offer more efficient and powerful representation capabilities for large-scale data.

In this context, the introduction of deep geometric SSMs aims to combine the representational power of deep neural networks with the interpretability and structured dynamics inherent in traditional SSMs. By incorporating the geometric properties of manifold-based modeling, these models adeptly capture the intrinsic structure of the data, which is crucial for accurately representing complex relationships in high-dimensional datasets, such as those found in brain imaging. This combination not only enhances interpretability but also allows for a more nuanced understanding of the underlying dynamics, ultimately improving the efficacy of the modeling process.

## A.2 Geodesic Distances Isometry

We aim to prove that the 'translation' operation defined by orthogonal transformations on the SPD manifold equipped with the Stein metric is an isometry. Specifically, we show:

$$d(\mathcal{T}_X(g), \mathcal{T}_Y(g)) = d(X, Y), \quad \forall X, Y \in \mathcal{M}, g \in \mathbb{O}. \tag{18}$$

Given the SPD manifold:

$$\mathcal{M} = \{X \in \mathbb{R}^{N \times N} \mid X = X^\top, X \succ 0\}, \tag{19}$$

the translation operation for orthogonal matrices $g \in \mathbb{O}$ (where $g^\top g = g g^\top = I$) is defined as:

$$\mathcal{T}_X(g) = gXg^\top. \tag{20}$$

The Stein metric is defined as:

$$d(X, Y) = \sqrt{\log \det \left( \frac{X + Y}{2} \right) - \frac{1}{2} \log \det(XY)}. \tag{21}$$

Orthogonal matrices preserve determinants:

$$\det(gXg^\top) = \det(X), \quad \text{since} \quad \det(g) = \pm 1. \tag{22}$$

We explicitly verify the invariance under orthogonal transformations:

$$d(\mathcal{T}_X(g), \mathcal{T}_Y(g)) = d(gXg^\top, gYg^\top)$$

$$= \sqrt{\log \det \left( \frac{gXg^\top + gYg^\top}{2} \right) - \frac{1}{2} \log \det \left( (gXg^\top)(gYg^\top) \right)}$$

$$= \sqrt{\log \det \left( g \frac{X + Y}{2} g^\top \right) - \frac{1}{2} \log \det \left( gXYg^\top \right)}$$

$$= \sqrt{\log \det \left( \frac{X + Y}{2} \right) + \log \det(g) + \log \det(g^\top) - \frac{1}{2} \left[ \log \det(XY) + \log \det(g) + \log \det(g^\top) \right]}.$$

Since $\det(g) = \pm 1$, we have $\log \det(g) = 0$. Thus,

$$d(\mathcal{T}_X(g), \mathcal{T}_Y(g)) = \sqrt{\log \det \left( \frac{X + Y}{2} \right) - \frac{1}{2} \log \det(XY)} = d(X, Y). \tag{23}$$

This proves the translation operation under orthogonal transformations is an isometry on the SPD manifold with the Stein metric.

## A.3 SPD Convolution Operation

*Proof.* Since $H$ is SPD, it can be decomposed as follows:

$$H = ZZ^\top, \tag{24}$$

where $Z = [z_1, z_2, \ldots, z_\theta]$ is a matrix of full rank. The convolutional result of an SPD representation matrix $X \in \mathbb{R}^{N \times N}$ can then be expressed as:

$$O = X * H = X * (ZZ^\top), \tag{25}$$

$$\Rightarrow X * (z_1 z_1^\top) + \cdots + X * (z_\theta z_\theta^\top), \tag{26}$$

$$\Rightarrow X * z_1 * z_1^\top + \cdots + X * z_\theta * z_\theta^\top, \tag{27}$$

where the transition from Eq. 26 to Eq. 27 uses the property of separable convolution. Suppose $z_i = [m_{i1}, m_{i2}, \ldots, m_{i\theta}]^\top$, for $i = 1, 2, \ldots, \theta$. The convolution between $X$ and $z_i$ can be written as:

$$X * z_i = P_{z_i} X, \quad X * z_i^\top = X P_{z_i}^\top, \tag{28}$$

where $P_{z_i} \in \mathbb{R}^{(M-N+1) \times M}$ and

$$P_{z_i} = \begin{bmatrix} m_{i1} & m_{i2} & \cdots & m_{iN} & 0 & 0 & \cdots \\ 0 & m_{i1} & m_{i2} & \cdots & m_{iN} & 0 & \cdots \\ 0 & 0 & m_{i1} & m_{i2} & \cdots & m_{iN} & \cdots \\ \vdots & \vdots & \vdots & \vdots & \vdots & \vdots & \vdots \\ 0 & 0 & \cdots & 0 & m_{i1} & m_{i2} & \cdots & m_{iN} \end{bmatrix}. \tag{29}$$

Thus, the following equations hold:

$$X * z_i * z_i^\top = P_{z_i} X P_{z_i}^\top, \tag{30}$$

and

$$O = X * Z = P_{z_1} X P_{z_1}^\top + \cdots + P_{z_\theta} X P_{z_\theta}^\top. \tag{31}$$

Since the rank of $P_{z_i}$ equals $M - N + 1$, the matrix $P_{z_i} X P_{z_i}^\top$ is also SPD. Therefore, for any $q \in \mathbb{R}^M$ where $q \neq 0$, we have:

$$q^\top O q = \sum_{i=1}^{\theta} q^\top P_{z_i} X P_{z_i}^\top q > 0. \tag{32}$$

Hence, $O$ is an SPD matrix.

Furthermore, the $k$-th channel of $X$ can be written as:

$$X^{(k)} = \sum_{l=1}^{L} X^{(l)} * H^{(k,l)}, \tag{33}$$

where $X^{(l)}$ denotes the $l$-th channel of the input descriptor. Since $X^{(l)}$ and $H^{(k,l)}$ are SPD matrices, and according to the above proof, $X^{(l)}$ is also an SPD matrix. Therefore, $X^{(k)}$ is a multi-channel SPD matrix.

## A.4 SPD $\exp(\cdot)$ Operation

*Proof:* Since $X$ is symmetric, we know that for any integer $k$, the powers $X^k$ are also symmetric. The matrix exponential of $X$ is defined by the following power series:

$$\exp(X) = \sum_{k=0}^{\infty} \frac{X^k}{k!}. \tag{34}$$

Each term in this series involves a symmetric matrix $X^k$, and the sum of symmetric matrices remains symmetric. Therefore, $\exp(X)$ is symmetric.

Since $X$ is symmetric, it can be diagonalized as: $X = Q\Lambda Q^\top$, where $Q$ is an orthogonal matrix (i.e., $Q^\top Q = I$) and $\Lambda$ is a diagonal matrix containing the eigenvalues $\lambda_1, \lambda_2, \ldots, \lambda_n$ of $X$. Because $X$ is positive definite, all eigenvalues $\lambda_i$ are positive, i.e., $\lambda_i > 0$ for all $i$.

The matrix exponential $\exp(X)$ is then given by:

$$\exp(X) = Q \exp(\Lambda) Q^\top, \tag{35}$$

where $\exp(\Lambda)$ is the diagonal matrix with entries $\exp(\lambda_1), \exp(\lambda_2), \ldots, \exp(\lambda_n)$. Since the exponential function satisfies $\exp(\lambda_i) > 0$ for all $\lambda_i \in \mathbb{R}$, each eigenvalue of $\exp(X)$ is positive. Thus, $\exp(X)$ has strictly positive eigenvalues, and since it is symmetric, it is also positive definite.

## A.5 SPD Padding $[\cdot]$ Operation

Given a SPD matrices $X \in Sym_N^+$ and a small positive value $\rho$, the assemble matrix $Y = diag(\rho, X) = \begin{bmatrix} \rho & 0 \\ 0 & X \end{bmatrix}$ is a SPD matrix.

*Proof:* First, $Y$ is a symmetric, since $Y^\top = \begin{bmatrix} \rho & 0 \\ 0 & X \end{bmatrix}^\top = \begin{bmatrix} \rho & 0 \\ 0 & X \end{bmatrix} = Y$. Then, to show that $Y$ is positive definite, we need to verify that for any non-zero vector $z = \begin{bmatrix} z_1 \\ z_2 \end{bmatrix} \in \mathbb{R}^{N+1}$, the quadratic form $z^\top Y z$ is strictly positive.

We compute the quadratic form:

$$z^\top Y z = \begin{bmatrix} z_1 & z_2^\top \end{bmatrix} \begin{bmatrix} \rho & 0 \\ 0 & X \end{bmatrix} \begin{bmatrix} z_1 \\ z_2 \end{bmatrix} = z_1^2 \rho + z_2^\top X z_2. \tag{36}$$

Since $\rho > 0$, the term $z_1^2 \rho \geq 0$, and it is strictly positive if $z_1 \neq 0$.

Furthermore, since $X \in Sym_N^+$, $X$ is positive definite, meaning $z_2^\top X z_2 > 0$ for any non-zero $z_2 \in \mathbb{R}^N$.

Thus, for any non-zero vector $z = \begin{bmatrix} z_1 \\ z_2 \end{bmatrix}$, we have:

$$z^\top Y z = z_1^2 \rho + z_2^\top X z_2 > 0. \tag{37}$$

which proves $Y \in Sym_{N+1}^+$ is a SPD matrix.

## A.6 Dataset

Table 4: The summarization of the HAR and HBC datasets.

| Dataset | # of sequences | # of classes | mean of lengths | # of joints/ROIs |
|---------|---------------|--------------|-----------------|------------------|
| UTKinect | 199 | 10 | 29 | 20 |
| Florece 3D Actions | 215 | 9 | 19 | 15 |
| HDM05 | 686 | 14 | 248 | 31 |
| HCP-WM | 17,296 | 8 | 39 | 360 |
| ADNI | 250 | 5 | 177 | 116 |
| OASIS | 1,247 | 2 | 390 | 160 |
| PPMI | 209 | 4 | 198 | 116 |
| ABIDE | 1,025 | 2 | 200 | 116 |

**For HAR dataset.** We evaluate the performance of the proposed *GeoDynamics* on three benchmark HAR datasets: the Florence 3D Actions dataset [61], the HDM05 database [53], and the UTKinect-Action3D (UTK) dataset [75]. The Florence 3D Actions dataset includes 9 activities (*answer phone, bow, clap, drink, read watch, sit down, stand up, tie lace, wave*), performed by 10 subjects, with each activity repeated 2 to 3 times, resulting in a total of 215 samples. These actions are represented by the motion of 15 skeletal joints. For the HDM05 dataset, we follow the protocol outlined in [71], selecting 14 action classes (*clap above head, deposit floor, elbow to knee, grab high, hop both legs, jog, kick forward, lie down on floor, rotate both arms backward, sit down chair, sneak, squat, stand up, throw basketball*). The sequences, captured using VICON cameras, result in 686 samples, each represented by 31 skeletal joints—significantly more than in the Florence dataset. The increased number of joints and higher intra-class variability make this dataset particularly challenging. Finally, the UTKinect-Action3D dataset consists of 10 action classes (*carry, clap hands, pick up, pull, push, sit down, stand up, throw, walk, wave hands*), captured using a stationary Microsoft Kinect camera. Each action was performed twice by 10 subjects, yielding 199 samples in total.

**For HBC dataset.** We select one dataset of healthy young adults and four disease-related human brain datasets for evaluation: the Human Connectome Project-Young Adult Working Memory (HCP-WM) [79], Alzheimer's Disease Neuroimaging Initiative (ADNI) [52], Open Access Series of Imaging

Studies (OASIS) [46], Parkinson's Progression Markers Initiative (PPMI) [50], and the Autism Brain Imaging Data Exchange (ABIDE). We selected a total of 1,081 subjects from the HCP-WM dataset. The working memory task included both 2-back and 0-back conditions, with stimuli featuring images of bodies, places, faces, and tools, interspersed with fixation periods. The specific task events are: 2bk-body, 0bk-face, 2bk-tool, 0bk-body, 0bk-place, 2bk-face, 0bk-tool, and 2bk-place. Brain activity was parcellated into 360 regions based on the multi-modal parcellation from [24]. For the OASIS (924 subjects) and ABIDE (1,025 subjects) datasets, which are binary-class datasets, one class represents a disease group and the other represent healthy controls. In the ADNI dataset, subjects are categorized based on clinical outcomes into distinct cognitive status groups: cognitively normal (CN), subjective memory concern (SMC), early-stage mild cognitive impairment (EMCI), late-stage mild cognitive impairment (LMCI), and Alzheimer's Disease (AD). For population analysis, we group CN, SMC, and EMCI into a "CN-like" group, while LMCI and AD form the "AD-like" group. This grouping enables a detailed analysis of cognitive decline and disease progression. The PPMI dataset consists of four classes: normal control, scans without evidence of dopaminergic deficit (SWEDD), prodromal Parkinson's disease, and Parkinson's disease (PD). This classification supports the study of different stages of Parkinson's progression. We employ Automated Anatomical Labeling (AAL) atlas [67] (116 brain regions) on ADNI, PPMI, ABIDE datasets, while Destrieux atlas [18] (160 brain regions) is used in OASIS to verify the scalability of the models.

### A.7 Comparison Methods and Experimental Results

We roughly summarize the comparison methods for HBC into two categories: spatial models and sequential models.

**Spatial models.** The spatial models are essential for understanding brain dynamics. Traditional GNNs like graph convolutional network (GCN) [44] and graph isomorphism network (GIN) [77] are selected for their ability to effectively capture diffusion patterns and isomorphism encoding in structured data. Subgraph-based GNNs, such as Moment-GNN [41], emphasize subgraph structures, enabling the identification of localized patterns that might be overlooked by traditional GNNs. Expressive GNNs, including graph substructure network (GSN) [7] and GNNAsKernel (GNN-AK) [81], are chosen for their enhanced expressivity through subgraph isomorphism counting and local subgraph encoding, which could be crucial for distinguishing subtle differences in complex systems.

A manifold-based model like the symmetric positive definite network (SPDNet) [16] is adopted for its ability to manage high-dimensional manifold data, making it suitable for more complicated datasets.

Two graph-based brain network analysis models for disease diagnosis, BrainGNN [49], an interpretable brain graph neural network for fMRI analysis, and ContrastPool [76], a contrastive dual-attention block and a differentiable graph pooling method.

Additionally, a traditional multi-layer perceptron (MLP) serves as a model due to its efficiency and versatility across various domains.

For all spatial models, following the optimal settings described in [60], we use the vectorized static functional connectivity (FC) as graph embeddings and the static FC matrices ($N \times N$) as adjacency matrices, where only the top 10% of edges are retained through thresholding to ensure sparsity. The input of SPDNet is the original $N \times N$ FC matrices.

For dynamic-FC models (STAGIN [43] and NeuroGraph [60], the thresholded dynamic FC matrices serve as the graph, NeuroGraph serve the vectorized FC as the embedding and STAGIN incorporates BOLD signals as part of the embedding, alongside its unique embedding construction method. For our *GeoDynamics*, we use the dynamic FC matrices as the input, resulting in $T \times N \times N$ matrices.

**Sequential models.** The sequential models are selected to analyze temporal dynamics in time-series BOLD signals. 1D-CNN is chosen for its ability to capture temporal patterns through convolutional operations. RNN [59] and LSTM [31] are included for their proficiency in modeling sequential data and capturing long-range dependencies. MLP-Mixer [65] is selected for its capability to mix both temporal and spatial features, offering a comprehensive view by integrating information across different dimensions. Transformer [68] is chosen for its powerful attention mechanisms, which allow it to capture global dependencies in sequential data. Brain network transformer (BNT) [40] is a tailored approach specifically designed for brain network analysis. Lastly, the state-space model

(SSM), represented by Mamba [25], is selected for its advanced state-space modeling abilities that effectively capture system dynamics over time.

For the sequential models, the inputs are the BOLD signals ($N \times T$).

Note, the inputs for all comparison methods align with the recent work presented in [21], ensuring fairness in the evaluation process.

We further conducted experiments using three brain network analysis models on disease-based datasets, including ADNI, OASIS, PPMI, and ABIDE. The diagnostic accuracies of 10-fold cross-validation are presented in Table 5. It is clear that our *GeoDynamics* consistently outperforms all the compared methods.

Table 5: Diagnostic accuracies on three popular brain network analysis models.

|  | ADNI | OASIS | PPMI | ABIDE |
|---|---|---|---|---|
| BrainGNN | 76.57 ± 10.01 | 86.07 ± 5.71 | 67.88 ± 10.32 | 62.24 ± 4.44 |
| BNT | 79.68 ± 6.15 | 86.07 ± 3.19 | 64.55 ± 16.80 | 69.99 ± 5.37 |
| ContrastPool | 80.08 ± 5.01 | 89.02 ± 4.22 | 69.78 ± 7.36 | 70.72 ± 3.45 |

## A.8 Hyperparameters, Inference time and Number of Parameters

We list the detailed hyperparameters of each model in Table 6. We summarize the inference time and the number of parameters of each mode on HCP-WM dataset ($N = 360, T = 39$) in Table 7, all the experiments are conducted on NVIDIA RTX 6000Ada GPUs. We can observe that our method efficiently utilized the parameters compared to most counterpart methods. Compared to Mamba (vanilla SSM), our method requires more time in the final step due to the logarithmic mapping, which involves the computationally expensive Singular Value Decomposition (SVD). However, it is more efficient than SPDNet (a manifold-based model), as we leverage convolution operations. As a result, the overall computational cost remains manageable.

Table 6: Hyperparameter settings for different models. $N$ denotes the number of brain regions. 'M-SGD' and 'M-Adam' represent Stochastic Gradient Descent (SGD) and Adam optimizers, respectively, equipped with manifold-aware updates that enforce geometric constraints (e.g., orthogonality).

| Model | 1D-CNN | RNN | LSTM | Mixer | TF | Mamba | GCN | GIN |
|---|---|---|---|---|---|---|---|---|
| **Optimizer** | Adam | Adam | Adam | Adam | Adam | Adam | Adam | Adam |
| **Learning rate** | $10^{-4}$ | $10^{-4}$ | $10^{-4}$ | $10^{-4}$ | $10^{-4}$ | $5 \times 10^{-5}$ | $10^{-4}$ | $10^{-4}$ |
| **Weight decay** | $5 \times 10^{-4}$ | $5 \times 10^{-4}$ | $5 \times 10^{-4}$ | $5 \times 10^{-4}$ | $5 \times 10^{-4}$ | $0$ | $5 \times 10^{-4}$ | $5 \times 10^{-4}$ |
| **Batch size** | 64 | 64 | 64 | 64 | 64 | 16 | 64 | 64 |
| **Epochs** | 300 | 300 | 300 | 300 | 300 | 300 | 300 | 300 |
| **Hidden dim** | 1024 | 1024 | 1024 | 1024 | 1024 | 1024 | 1024 | 1024 |
| **Network layer** | 2 | 2 | 2 | 4 | 4 | 4 | 2 | 2 |
| **Model** | GSN | MGNN | GNN-AK | SPDNet | MLP | STAGIN | NeuroGraph | *GeoDynamics* |
| **Optimizer** | Adam | Adam | Adam | M-SGD | Adam | Adam | Adam | M-Adam |
| **Learning rate** | $10^{-2}$ | $10^{-2}$ | $10^{-3}$ | $5 \times 10^{-3}$ | $10^{-4}$ | $5 \times 10^{-4}$ | $10^{-4}$ | $5 \times 10^{-5}$ |
| **Weight decay** | $0$ | $5 \times 10^{-4}$ | $0$ | $10^{-5}$ | $5 \times 10^{-4}$ | $10^{-3}$ | $5 \times 10^{-4}$ | $0$ |
| **Batch size** | 16 | 16 | 128 | 32 | 64 | 3 | 16 | 16 |
| **Epochs** | 300 | 1000 | 100 | 100 | 300 | 100 | 100 | 300 |
| **Hidden dim** | 256 | 1024 | 128 | $[N, 64, 32]$ | 1024 | 128 | 32 | $N$ |
| **Network layer** | 2 | 2 | 2 | 3 | 2 | 4 | 3 | 2 |

Table 7: Model inference time (ms/item) and the number of parameters (M) comparison across various architectures on HCP-WM dataset.

|  | GCN | GIN | GSN | MGNN | GNN-AK | SPDNet | MLP | 1D-CNN |
|---|---|---|---|---|---|---|---|---|
| Time (ms) | 2.29 | 2.28 | 3.40 | 2.23 | 38.18 | 27.05 | 2.67 | 0.93 |
| Para (M) | 1.79 | 3.89 | 0.92 | 4.94 | 290.3 | 0.19 | 66.9 | 2.22 |
|  | RNN | LSTM | Mixer | TF | Mamba | NeuroGraph | STAGIN | *GeoDynamics* |
| Time (ms) | 0.87 | 0.91 | 0.91 | 1.21 | 0.33 | 39.79 | 20.92 | 2.51 |
| Para (M) | 1.19 | 14.45 | 6.78 | 12.98 | 27.05 | 0.29 | 1.17 | 14.60 |

## A.9 Discussion

We expect our manifold-based deep model to facilitate our understanding on brain behavior in the following ways.

*(1) Enhance the prediction accuracy.* A plethora of neuroscience findings indicate that fluctuation of functional connectivities exhibits self-organized spatial-temporal patterns. Following this notion, we conceptualize that well-defined mathematical modeling of intrinsic data geometry of evolving functional connectivity (FC) matrices might be the gateway to enhance prediction accuracy. Our experiments have shown that respecting the intrinsic data geometry in method development leads to significantly higher prediction accuracy for cognitive states, as demonstrated in Fig. 3.

*(2) Enhance the model explainability.* We train the deep model to parameterize the transition of FC matrices on the Riemannian manifold (Eq. 14). By doing so, we are able to analyze the temporal behaviors with respect to each cognitive state using post-hoc complex system approaches such as dynamic mode decomposition, stability analysis.

*(3) Provide a high-order geometric attention mechanism that is beyond node-wise or link-wise focal patterns.* Conventional methods often employ attention components for each region or link in the brain network separately, thus lacking the high-order attention maps associated with neural circuits (i.e., a set of links representing a sub-network). In contrast, the SPD-preserving attention mechanism (Eq. 16) in our method operates on the Riemannian manifold, taking the entire brain network into account. As shown in Fig. 4, our method has identified not only links but also sub-networks relevant to cognitive states and disease outcomes.

## A.10 Limitations and Future Work

First, the use of matrix exponentials and logarithmic operations (core to our Riemannian framework), requires eigenvalue or Cholesky decompositions, which are computationally intensive. This leads to higher complexity compared to standard Euclidean approaches, especially as the number of nodes (i.e., brain regions) increases. To address this challenge, we can leverage parallel computing to mitigate the computational burden.

Second, the effectiveness of manifold-based modeling depends on the assumption that input SPD matrices lie within a well-behaved region of the manifold (e.g., within a geodesic ball), which ensures the uniqueness and stability of the Fréchet mean. When this assumption is violated, such as in highly noisy or poorly conditioned data (rank-deficient), the performance of the method may degrade.

Third, interpretability remains a challenge. While the geometric framework captures intrinsic data structure more faithfully, understanding how specific patterns relate to clinical or cognitive outcomes is still an open area of research.

In future work, we plan to: (1) Optimize runtime further through low-rank approximations or manifold-aware pruning. (2) Extend the framework to handle multi-modal neuroimaging data (e.g., EEG, MEG). (3) Explore other interpretable models to bridge the gap between mathematical complexity and clinical insight.

## A.11 Impact Statement

This work bridges the fields of machine learning and neuroscience by introducing a geometric deep learning-based state-space model (*GeoDynamics*) for understanding brain dynamics and their relationship to behavior and cognition. By leveraging the unique properties of Riemannian geometry, our model offers a holistic view of brain state evolution as a self-organized dynamical system, addressing critical challenges in functional neuroimaging studies and enabling applications in early diagnosis of neurodegenerative diseases such as Alzheimer's and Parkinson's.

Beyond neuroscience, *GeoDynamics* demonstrates broad applicability in capturing spatio-temporal dynamics across diverse domains, as evidenced by its promising performance in human action recognition benchmarks. This highlights the potential of our approach to impact fields ranging from healthcare to computer vision, offering tools for scalable, robust, and interpretable analysis of complex systems.

