# OpenReview forum: "GeoDynamics: A Geometric State‑Space Neural Network for Understanding Brain Dynamics on Riemannian Manifolds"
_NeurIPS.cc/2025/Conference — NeurIPS 2025 poster_

### Official Review · Reviewer_X13L · 2025-06-26

**Clarity:** 2
**Significance:** 3
**Originality:** 3
**Rating:** 5
**Confidence:** 4

**Summary:**

This paper introduces a neural network based on a state-space model that operates on the Riemannian manifold of SPD matrices. The proposed network is tailored for processing sequences of functional connectivity matrices, which are modeled as SPD matrices that capture the dynamics of brain activity. The proposed network is evaluated on various benchmarks involving brain dynamics and action recognition.

**Questions:**

Specific comments and suggestions:
- Line 74: specify which “Riemannian manifold”
- Line 80: “manifold instance” => sample/point on the manifold
- Sec. 2.2 - “Riemannian geometry algebra” => “Riemannian geometry”
- Sec. 2.2, line 120: the notation in parenthesis is not clear. My suggestion is to replace \mathcal{M} => sym^+_N.
- Sec. 2.2 - only the Stein metric (aka Jensen-Bregman logdet divergence) [10] is mentioned and used but there are many other Riemanian metrics on the SPD manifold, which have been proposed and studied extensively. It is important to mention the alternatives, e.g., the affine-invariant metric [*] and log Euclidean [**], and discuss why the particular logdet divergence was chosen.

[*] Pennec, X., Fillard, P., & Ayache, N. (2006). A Riemannian framework for tensor computing. International Journal of computer vision, 66, 41-66.

[**] Arsigny, V., Fillard, P., Pennec, X., & Ayache, N. (2006). Log‐Euclidean metrics for fast and simple calculus on diffusion tensors. Magnetic Resonance in Medicine: An Official Journal of the International Society for Magnetic Resonance in Medicine, 56(2), 411-421.

**Ethical Concerns:**

["NO or VERY MINOR ethics concerns only"]

**Final Justification:**

The authors’ response adequately addresses my concerns, particularly regarding the lack of detail, the treatment of Riemannian geometry, the choice of metric, and the discussion of limitations. Provided that all these changes are incorporated into the revised paper, and after considering the other reviews and responses, I have decided to raise my score to 5: accept.

**Limitations:**

Some limitations are mentioned in the appendix (A.11) but I find it quite hidden, and some of the discussion could be moved to the main text

**Quality:**

3

**Strengths And Weaknesses:**

Strengths:
- The addressed problems in neuroscience are important and central
- The proposed network is new and nicely combines concepts and tools from different fields, where the internal state and output updates are performed “on the manifold” using translation and the weighted Riemannian mean (Eq. 3)
- The experiments demonstrate the benefits of the proposed network compared to baselines in a very broad range of applications

Weaknesses:
- The presentation and writing need some improvement, specifically, the treatment of the Riemannian geometry of SPD matrices (see details below)
- Implementations details are missing - for example, details on how the Frechet mean is computed
- The computational complexity is mentioned only in Appendix A.11 in the context of limitations without details. From practical standpoint, this is an important issue that should be better discussed. Specifically, what is the computational complexity of the algorithm and how does it scale with the size of the FC matrices?

---

> ### Author Rebuttal · Authors · 2025-07-30
>
> **We sincerely thank the reviewer for their positive and constructive feedback. We are glad that the reviewer recognizes the importance of the problems we address in neuroscience, as well as the novelty of our neural network model based on state-space dynamics on the SPD manifold. We appreciate the encouraging comments on how our method combines tools from different fields and performs intrinsic updates on the manifold. We also thank the reviewer for pointing out areas for improvement. In the following, we respond to each concern and suggestion in detail.**
>
>
> **Q1: The presentation and writing need some improvement, specifically, the treatment of the Riemannian geometry of SPD matrices. (see details above)**
>
> **A1:** Thank you for your detailed and insightful comments regarding the use and presentation of Riemannian geometry on the SPD manifold. We have carefully revised the manuscript to address the clarity and terminology issues you highlighted:
>
> - **Line 74**: We now explicitly specify the manifold as the space of $N \times N$ symmetric positive definite (SPD) matrices, denoted $\text{Sym}^+_N$, a well-studied Riemannian manifold in information geometry and medical imaging.
>
> - **Line 80:** We have replaced “manifold instance” with “a point on the manifold” to align with standard terminology in differential geometry.
>
> - **Section 2.2:** We have corrected “Riemannian geometry algebra” to “Riemannian geometry” throughout. In line 120, we now clarify the notation by replacing $\mathcal{M}$ with $\text{Sym}^+_N$ to improve precision.
>
> - **Metric choice**: Thank you for your insightful comment. We agree that many Riemannian metrics have been proposed for the SPD manifold, including the affine-invariant Riemannian metric (AIRM) [Pennec et al., 2006] and the log-Euclidean metric [Arsigny et al., 2006]. In the revised manuscript (Sec. 2.2), we have included a discussion of these alternatives and clarified our rationale for selecting the Stein metric (also known as Jensen-Bregman log-det divergence).
>
> Compared to AIRM and log-Euclidean, the Stein metric offers several benefits for our application:
>
> ○ Like AIRM and log-Euclidean, it respects the intrinsic Riemannian geometry of SPD matrices.
>
> ○ It is invariant under congruent transformations, which is important for cross-subject comparisons in functional connectivity.
>
> ○ It avoids the computational cost of matrix logarithms and square roots.
>
> To empirically support this choice, we conducted an ablation study on the ADNI dataset, comparing classification accuracy and runtime across three manifold-based metrics:
>
>
> | Metric        | Formulation                                                                                                         | Accuracy (%) | Runtime (ms) |
> | ------------- | ------------------------------------------------------------------------------------------------------------------- | ------------ | ------------ |
> | AIRM          | $\|\log(\mathbf{X}^{-1/2} \mathbf{Y} \mathbf{X}^{-1/2})\|\_F\$                                                      | 81.51        | 1.525        |
> | Log-Euclidean | $\|\log(\mathbf{X}) - \log(\mathbf{Y})\|\_F\$                                                                       | 83.10        | 1.488        |
> | Stein (ours)  | \$\sqrt{ \log\det\left( \frac{\mathbf{X} + \mathbf{Y}}{2} \right) - \frac{1}{2} \log\det(\mathbf{X} \mathbf{Y}) }\$ | **85.49**    | **1.375**    |
>
> As shown above, the Stein metric achieves the best trade-off between classification performance and computational efficiency. These findings, combined with its theoretical strengths, support our decision to adopt the Stein metric as the default geometry-aware distance measure in our model. The manuscript has been updated to reflect this comparison and justification.
>
> These clarifications and revisions have been incorporated into Section 2.2 and we believe they improve both the rigor and readability of the presentation. Thank you again for your constructive feedback.
>
> **Q2: Implementations details are missing - for example, details on how the Frechet mean is computed**
>
> **A2:** Thank you for your insightful comment. We now provide a detailed explanation of how the weighted Fréchet mean is computed in our implementation:
> 1. $L$ is initialized as the identity matrix so that $F=L L^{\top}+\varepsilon I$ remains SPD. (the complexity is $O(N^3)$)
> 2. The algorithm minimizes the weighted sum of squared LogDet distances over the SPD matrices in $X$​ by updating $L$ using an optimizer (Adam). This differentiates the log-determinant functions concerning $F$ while respecting the SPD manifold step typically involves structure. (the complexity is $O(N^3)$)
> 3. The process stops when the change in loss falls below a given tolerance, ensuring convergence. (the complexity is $O(N^3)$)
>
> In addition, the translation is defined as: $\mathcal{T}(X,g):=g X g^{\top}$. The implementation details are provided below.
>
> 1. Reshapes vector $g$ as a row vector;
>
> 2. From the row vector, a lower triangular matrix $L$ is constructed by iterating over each row, extracting the required slice of $g$, and padding with zeros; (need interaing N rows, the complexity is $O(N^2)$);
>
> 3. A skew-symmetric matrix is formed by subtracting the transpose of the lower triangular matrix: $G_{skew}​= L-L^⊤$;
>
> 4. The skew-symmetric matrix is then mapped into a rotation matrix in $\mathbb O(n)$ using the matrix exponential: $G=exp(G_{skew})$; (involves inverting and multiplying operation, so the complexity is $O(N^3)$).
>
> 5. The translated SPD matrix is computed by conjugating $X$ with $G$: $\mathcal{T}(X,G):=G X G^{\top}$. (including two matrix multiplications, the complexity is $O(N^3)$).
>
> We will incorporate the implementation details in the final version, thank you so much.
>
> **Q3: The computational complexity is mentioned only in Appendix A.11 in the context of limitations without details. From a practical standpoint, this is an important issue that should be better discussed. Specifically, what is the computational complexity of the algorithm and how does it scale with the size of the FC matrices?**
>
> **A3:** Thank you for this constructive comment. If the FC matrices are $N\times N$, and both the input $X$ and the convolution kernel $K$ are $N\times N$, then each output element requires an inner product over $N^2$ elements. Since the convolution is computed at each spatial location—amounting to $N^2$ output elements—the overall time complexity for one time step is $O(N^4)$. When considering  $T$ time points, the total complexity becomes: $O(T \cdot N^4)$. nn.DataParallel in PyTorch can achieve parallel computing, which distributes the computation across multiple GPUs. We have included the inference time in Table 5 and will add this analysis in the final version.
>
> **Q4: Some limitations are mentioned in the appendix (A.11) but I find it quite hidden, and some of the discussion could be moved to the main text**
>
> **A4:** Thank you for your helpful suggestion. It should be an easy fix by highlighting the existing write-ups. In the revised version, we will move a concise summary of the key limitations to the main text (end of Section 4.3), including aspects such as computational cost, assumptions about manifold structure, and interpretability challenges. We will retain the detailed discussion in the appendix for completeness but make cross-references more prominent to ensure these important considerations are accessible to readers.
>
> **Thanks again for your feedback, if the reviewer has additional questions, we would be glad to provide further clarifications.**

---

> > ### Comment · Reviewer_X13L · 2025-08-05
> >
> > I would like to thank the authors for their detailed response. It addresses my concerns, and I encourage the authors to incorporate all these changes into the revised paper. After reading the other reviews and responses, I have no further comments and have decided to raise my score.

---

> > > ### Author Response · Authors · 2025-08-05
> > >
> > > We sincerely appreciate it. We are committed to incorporating all the changes into the final version.

---

### Official Review · Reviewer_bcvk · 2025-07-02

**Clarity:** 3
**Significance:** 3
**Originality:** 3
**Rating:** 5
**Confidence:** 3

**Summary:**

The goal of this paper is to analyze functional MRI (fMRI) data to study human brain with respect to clinical conditions or some cognitive tasks being performed by the patient. There are two claimed innovations in this paper. Firstly, it uses a state-space model on the SPDM manifold to model the times series data. Specifically, the approach is to transform the data into a time series of brain connectivity matrices, and to model it as a time series on SPDM manifold. To analyze this time series, the paper formulates a Markov decision process that models a hidden state, a transition matrix, and an output that classifies the state. The paper user geometry of SPDM to formulate these elements intrinsically on the manifold. Secondly, the inference is performed using a neural-network architecture. It defines a convolution operation on SPDMs for use in forming convolution layers in the network.

There is a very extensive experimental support to the approach in the paper. It compares several methods on datasets for classifying cognitive tasks and neurodegenerative diseases from fMRI data. The results are generally superior to the existing methods.

**Questions:**

Please clarify the novelty of your approach to the part papers on dynamic brain functional connectivity.

**Ethical Concerns:**

["NO or VERY MINOR ethics concerns only"]

**Final Justification:**

In view of the discussion with the authors and their response to my comments, I retain my original (very high) rating of this paper.

**Limitations:**

I don't see any note on the limitations of the model/method proposed here.

**Paper Formatting Concerns:**

None noted.

**Quality:**

3

**Strengths And Weaknesses:**

Strengths
1.	It utilizes the geometry of SPDMs to respect the structure of brain connectivity. Several papers have utilized this representation to constrain analysis to interpretable solutions.
2.	It implements a deep-learning version of the state space models on nonlinear manifolds.
3.	The experimental results are both extensive and impressive.

Weaknesses:

1.	The modeling of brain functional connectivity using a time-series of SPD matrices is not new, a number of papers have used this formulation. See for example, Dai et al, Analyzing Dynamical Brain Connectivity as Trajectories on Space of Covariance Matrices, 2020. Implementing this SSM using neural networks has some novelty to it.
2.	The computational cost of performing analysis on SPDM manifold is expensive. The matrix computations grow much faster than the size of matrices. The method will not scale as well if the number of nodes in the brain increases.
3.	Even though the empirical results are impressive, this formulation is hard to analyze theoretically due to nonlinearity of the underlying manifold.

---

> ### Author Rebuttal · Authors · 2025-07-30
>
> **We sincerely appreciate the reviewers' constructive feedback and thoughtful critiques. Below, we provide point-by-point responses to address each concern and outline the improvements made in the revised manuscript.**
>
> **Q1: Please clarify the novelty of your approach to the part papers on dynamic brain functional connectivity.**
>
> **A1** Thank you for pointing this out. We fully agree that modeling brain functional connectivity as a trajectory of SPD matrices has been explored in prior work, such as Dai et al. (2020). Our contribution builds upon this foundation but introduces a novel integration of *geometric deep learning and state space modeling (SSM)* within the Riemannian manifold framework.
>
> Specifically, our approach differs in three important ways (Line 71-94 in the original manuscript):
>
> - **Structural novelty**: We formulate dynamic FC modeling as a learning task on the Riemannian manifold of SPD matrices, treating each FC matrix as a point on the manifold and encoding whole-brain interactions more holistically than Euclidean approximations.
>
>
> - **Behavioral modeling via SSM**: Unlike prior approaches that either lack temporal modeling or rely on RNNs, we adopt a continuous-time state space model (SSM) to capture latent neural dynamics. This dual formulation of state and observation equations enables a principled mapping between brain states and observed neural signals over time.
>
>
> - **Geometric–neural integration**: We replace traditional Euclidean algebra in SSMs with Riemannian geometry, enhancing the model's ability to handle the curvature and structure of SPD space. This is paired with a geometric-adaptive attention mechanism that improves efficiency and generalization (Sec. 3).
>
> While Dai et al. focused on geometry-aware trajectory analysis (a valuable contribution), our work adds a temporal modeling layer via SSM and neural parametrization to learn both latent dynamics and manifold structure in an end-to-end fashion. To our knowledge, our work is the first framework to unify these components within a deep geometric setting for functional brain analysis.
>
> We have validated our method not only on multiple brain datasets (e.g., ADNI, OASIS, HCP, PPMI, ABIDE), but also on human action recognition tasks, demonstrating generalizability across domains. We will make more clearer in the final version.
>
> **Q2: The computational cost of performing analysis on SPDM manifold is expensive. The matrix computations grow much faster than the size of the matrices. The method will not scale as well if the number of nodes in the brain increases.**
>
> **A2:** Thank you for highlighting this important point. We acknowledge that operations on the SPD manifold—especially matrix logarithms and exponentials—can become computationally expensive as the number of brain regions increases (we have discussed this limitation (Appendix A.11-Page 22) in the original manuscript). This is a general challenge for manifold-based methods. To address this, we currently leverage **parallel computing and batched matrix operations** to significantly reduce the computational burden during training and inference. Additionally, since many SPD operations are independent across time steps or subjects, they are naturally suited to GPU acceleration and distributed computing.
>
> Looking ahead, we plan to explore more **scalable approximations**, such as low-rank factorizations, tangent-space projections, or spectral truncation of covariance matrices, which can maintain performance while further improving efficiency.
> We appreciate the reviewer’s comment and will clarify these scalability considerations in the revised manuscript.
>
> **Q3: Even though the empirical results are impressive, this formulation is hard to analyze theoretically due to nonlinearity of the underlying manifold.**
>
> **A3:** Thank you for raising this insightful point. We agree that theoretical analysis becomes more complex when working with nonlinear Riemannian manifolds such as the space of SPD matrices. This is a known limitation of many geometry-based deep models.
>
> To mitigate this challenge, our framework is carefully designed to preserve theoretical tractability where possible. Specifically:
>
> - We **replace Euclidean algebra in state space models with Riemannian counterparts** in a principled way, allowing certain analogs of stability and controllability theory to still be studied in the manifold setting (see Section 3).
>
> - The **structure of the SSM** provides an interpretable framework, with latent dynamics governed by well-defined ODEs and output mappings, which serve as a bridge between complex geometry and system theory.
>
> - For practical robustness, we **regularize manifold operations** using exponential map truncation and intrinsic norms, which helps constrain model behavior and improve empirical generalization.
>
> We also note that the field is actively developing new theoretical tools for analyzing learning dynamics on manifolds (e.g., Riemannian optimization and tangent space approximations). We believe our work contributes to this ongoing effort by demonstrating a successful application in the neuroscience domain, and future work will aim to formalize the theoretical underpinnings further.
>
> We will clarify this discussion in the revised manuscript.
>
> **Q4: I don't see any note on the limitations of the model/method proposed here.**
>
> **A4:**  Thank you for your important observation. Due to space constraints in the main text, we have included a detailed discussion of the model’s limitations in **Appendix A.11 (Page 22)** of the original manuscript. This section outlines key limitations, such as the computational overhead introduced by Riemannian operations and challenges related to noisy or rank-deficient input data.  We will make this cross-reference more prominent in the revised version to ensure readers are aware of these important considerations. Thank you so much.

---

> > ### Comment · Reviewer_bcvk · 2025-08-05
> > **Comment on Rebuttal**
> >
> > Thank you very much for a detailed response on my comments and questions. I agree with most of the claims about novelty although I am less inclined to rate them as transformative. I believe that this paper brings together several of the current ideas, implements them using neural networks, and demonstrate these ideas convincingly using several datasets. With that said, I am very supportive of accepting this paper and will retain my original rating.

---

> > > ### Author Response · Authors · 2025-08-05
> > >
> > > Thank you very much for your thoughtful feedback throughout the review process. We sincerely appreciate your recognition of the novelty and the overall contribution of our work.

---

### Official Review · Reviewer_nqTq · 2025-07-03

**Clarity:** 1
**Significance:** 2
**Originality:** 3
**Rating:** 4
**Confidence:** 4

**Summary:**

This paper introduces GeoDynamics, a geometric state-space neural network designed to model time-varying functional connectivity (FC) in the brain. Unlike standard SSMs which operate in Euclidean space, GeoDynamics constrains both system inputs, hidden states, and outputs to lie on the Symmetric Positive Definite (SPD) manifold, motivated by the observation that brain FC matrices are naturally SPD. The transition dynamics are defined using group actions (particularly of the orthogonal group) and Riemannian constructs like the Fréchet mean and the Stein metric. The model is evaluated on both human brain datasets and a human action recognition benchmark.

**Questions:**

1. Model components: Could you explicitly define what constitutes the input $X(t)$, the hidden state $S(t)$, and the observed output $Y(t)$ in the context of your neuroscience application? It is currently unclear how the functional connectivity matrices relate to these quantities.
2. Equation 3 confusion: What is the set over which you are computing the Fréchet mean in Equation 3? $S^{(k-1)}$ appears to be a single SPD matrix, not a set. What is $\tilde{A}$? Ostensibly it is a set of weights (scalars).
3. Group action inconsistency: You define the translation $\mathcal{T}$ using the action of the orthogonal group $\mathbb{O}$ on the SPD manifold, but then apply it to the Fréchet mean of SPD matrices. If the argument to $\mathcal{T}$ must be orthogonal, this would be undefined unless the mean is trivially the identity. Can you clarify?
4. On Fréchet mean and uniqueness: What are the guarantees on the existence and uniqueness of the Fréchet mean under your assumptions? Do you enforce closeness on the manifold during training?
5. Experimental focus: Why include human action recognition? How does this task align with your main objective of modeling brain dynamics? Could the model be evaluated more thoroughly on neuroscience-relevant tasks?

My score could increase if the authors:

- Sharply revise the manuscript to clarify definitions and model structure,
- Cleanly separate results from interpretation,
- Remove redundant text and clarify inconsistent terminology,
- Provide more focused experimental evaluation and justification for each task.

**Ethical Concerns:**

["NO or VERY MINOR ethics concerns only"]

**Final Justification:**

I had major concerns about the clarity and readability of the manuscript, especially unclear mathematical notation. The authors have provided significant clarification in their rebuttal, and made a commitment to increase the clarity and focus of their manuscript.

They have also conducted additional ablation experiments and benchmarks.

Therefore, I have increased my initial score (2) to a (4).

**Limitations:**

Yes

**Quality:**

1

**Strengths And Weaknesses:**

### Strengths

- Principled use of Riemannian geometry: The authors leverage interesting manifold-aware constructs (e.g., the Fréchet mean, group actions, Stein metric) to build the model in a way that respects the geometry of SPD matrices.
- Theoretical grounding: There are rigorous definitions and group-theoretic justifications, and the paper includes formal results and claims with proofs in the appendix.
- Potentially generalizable: While aimed at brain data, the method could be extended to any domain where SPD-valued time series arise.

### Weaknesses

- Major clarity issues throughout:
    - Core concepts such as system input, hidden state, and output are not clearly mapped to the neuroscience application (e.g., is the FC the input or the hidden state?).
    - Equations (e.g., Equation 3) are difficult to parse and are not explained with sufficient care.
    - Basic elements of the SSM formalism (e.g., matrices A, B, C, D) are not introduced for readers unfamiliar with control theory.
    - Redundant or imprecise definitions (e.g., repeated definitions of group actions) obscure the main argument.
- Lack of focus in narrative: The inclusion of the human action recognition dataset seems like an afterthought and dilutes the focus of the paper, which is ostensibly about modeling brain dynamics.
- Experimental presentation is weak: The structure of the experimental section is muddled. Results are intermixed with speculative interpretation (”Remarks”), and important clarifications (e.g., what is being predicted, what the task is) are missing.
- Terminological inconsistency: Terms like "system state" vs. "hidden state" are used interchangeably, it would help to stick to one of them.
- Foundational confusion: Some parts of the model formulation appear self-inconsistent (e.g., the domain of the group action $\mathcal{T}$ vs. its usage in Equation 3), and the writing does not help to disambiguate.

---

> ### Author Rebuttal · Authors · 2025-07-30
>
> **We appreciate your recognition of our method's theoretical grounding and geometric formulation. Below we respond to each of your concerns and will incorporate all suggestions into the final version.**
>
>
> **Q1: Core concepts such as system input, hidden state, and output are not clearly mapped to the neuroscience application (e.g., is the FC the input or the hidden state?)**
>
> **A1:** Sorry for the unclear description. The FC matrices $X(t)$ are treated as the system input in our model. The hidden state $S(t)$ represents the internal geometric representation on the manifold (learned from $X(t)$ that summarizes historical FC dynamics, and the system output $Y(t)$ corresponds to the predicted FC state used for downstream inference tasks (e.g., brain state identification). We have updated **Sec. 3.1-Overview** to clearly define the mapping between the neuroscience data and the model variables. These changes will be reflected in the revised manuscript to improve clarity and align the mathematical formulation with the neuroscience context.
>
> **Q2: Equations (e.g., Equation 3) are difficult to parse and are not explained with sufficient care**
>
> **A2:** Eq. 3 is derived as the manifold-based extension of Eq. 1. The key distinction lies in the underlying geometry of the system:
> - Eq. 1 corresponds to a vanilla SSM defined in Euclidean space, where system evolution and observations are governed by standard linear operations.
> - Eq. 3 reformulates the SSM on the Riemannian manifold of symmetric positive definite (SPD) matrices, where FC matrices are naturally embedded.
>
> In this manifold setting, standard operations (e.g., averaging, updating states) must be replaced with geometrically consistent operations, including:
>
> - the Fréchet mean $\mathcal{F}$,  which generalizes the concept of averaging to Riemannian spaces, and
> - the translation operator $\mathcal{T}$, which allows us to evolve system states and outputs across the manifold.
>
> We will revise Sec. 3.1 to walk readers through these concepts more carefully and to explain each term in Eq. 3 with explicit reference to their role in modeling the time-varying FC matrices on the SPD manifold.
>
> **Q3: Basic elements of the SSM formalism (e.g., matrices A, B, C, D) are not introduced for readers unfamiliar with control theory.**
>
> **A3:** We agree that clarity around the role of A,B,C,D is important, especially for readers unfamiliar with control theory. Fortunately, it should be an easy fix in the final version by highlighting the existing write-ups. As noted in Sec. 2.1 (line 110-117), we have already introduced the standard linear state-space model and defined each matrix. In our proposed model, these parameters retain similar functional roles but operate on SPD manifolds through geometry-aware operations (e.g., Fréchet mean and manifold translation), as described in Sec. 3.1 and Eq. (3). We will make this connection between the classical and manifold-based formulations more explicit in the revised version.
>
> **Q4: About group action definitions**
>
> **A4:** Thank you for these insightful comments. We have unified and clarified our definitions in Section 2.2 as follows:
> - 1. **Group action on SPD matrices:**
> $ g \cdot X := gXg^\top,
>    \quad g \in \mathbb{G},; X \in \mathcal M.$
>    Here, $\mathbb{G}$ denotes the **general linear group** of full‑rank $N\times N$ matrices.
> - 2. **Restriction to orthogonal subgroup:**
>    We then restrict to the **orthogonal subgroup**
>    $\mathbb{O}\subset\mathbb{G}$, consisting of all $N\times N$ orthogonal matrices that preserve the Stein metric.
> - 3. **Translation operator:**
>  $
>    \mathcal T: \mathcal M \times \mathbb{O} \to \mathcal M,
>    \quad
>    \mathcal T(X,g) = gXg^\top.
> $
>    In this definition, the first argument $X$ may be *any* SPD matrix (e.g., the output of the Fréchet mean), and the second argument $g$ is an orthogonal matrix.
>
> - 4. **Application in Eq. (3):**
> $S^{(k)} = \mathcal T\bigl(\mathcal F(\cdots),,\widetilde A\bigr),$
>    which denotes the action of the orthogonal matrix $\widetilde A \in \mathbb{O}$ on the SPD output of $\mathcal F$.  Since $\mathcal F(\cdot)$ always returns an SPD matrix, $\mathcal T(\mathcal F(\cdot),\widetilde A)$ is well‑defined.
>
> We have updated Lines 126–133 accordingly to remove any redundant or unnecessary repetitions. Thanks.
>
> **Q5: Equation 3 confusion: What is the set over which you are computing the Fréchet mean in Equation 3? S^(k-1) appears to be a single SPD matrix, not a set. What is  Ostensibly it is a set of weights (scalars).**
>
> **A5:** Thank you for this insightful question. We appreciate the opportunity to clarify Eq. 3, particularly the structure and interpretation of the weighted Fréchet mean.  In Eq. 3 the Fréchet mean is not taken over a singleton {$S^{(k-1)}$}, but over a sliding window of past states (and inputs) with their associated weights. Concretely, letting the window length be $\tau$, we compute
> $$
> S^{(k)} = \mathcal{T} \left(
> \mathcal{F}_1(\cdot), \,
> \mathcal{F}_2(\cdot)
> \right)
> $$
>
> $\mathcal{F}_1(\cdot) = \mathcal{F}\left(
> \left\{ S_j \right\}_{j=k-\tau}^{k-1},
> \left\{ A_j \right\}_{j=k-\tau}^{k-1}
> \right)$
>
> $\mathcal{F}_2(\cdot) = \mathcal{F}\left(
> \left\{ X_j \right\}_{j=k-\tau}^{k},
> \left\{ B_j \right\}_{j=k-\tau}^{k}
> \right)$
>
> The full expression is too long to display clearly in a single line, so we break it down into individual components for clarity:
>
> | Fréchet mean  | First Argument                         | Second Argument                        |
> |-----------------|----------------------------------------|----------------------------------------|
> | $\mathcal{F}_1(\cdot)$ |  {$S_j$} $_{j={k-\tau}}^{k-1}$            |   {$A_j$} $_{j={k-\tau}}^{k-1}$            |
> | $\mathcal{F}_2(\cdot)$ |  {$X_j$} $_{j={k-\tau}}^{k}$              |   {$B_j$} $_{j={k-\tau}}^{k}$              |
>
>
>  so that the set  {$S_j$} $_{j={k-\tau}}^{k-1}$ contains the $\tau$ most‑recent system–state SPD matrices (window size), with transition matrix $\{A_j\}$,
>
> the set  {$X_j$} $_{j={k-\tau}}^{k}$ contains the same window of inputs plus the current $X^k$. We have updated Eq. 3 and the related text to make this explicit. We have also conducted an ablation study in terms of difference window size (Appendix A.9 in the original manuscript). We will make it more clear in the final version, thanks.
>
>
> **Q6: What are the guarantees on the existence and uniqueness of the Fréchet mean under your assumptions? Do you enforce closeness on the manifold during training?**
>
> **A6:** Under the affine‑invariant metric, SPD(n) is a Hadamard manifold (complete, simply‑connected, non‑positive curvature) [Afsari, B. (2011). Riemannian $𝐿^𝑝$ center of mass: existence, uniqueness, and convexity. Proceedings of the American Mathematical Society]. Hence, there exists a unique Fréchet mean given a set of FC matrices. We initialize each translation matrix $g$ close to the identity (so $g \thickapprox I$) and train with small learning rates using our manifold‑aware optimizers (M‑Adam) (Table 5 in the original manuscript). This guarantees that each update exp(Δ) moves only a short geodesic distance on $\mathbb O(n)$, keeping all points for the Fréchet mean within a convex neighborhood.
>
> **Q7: Could the model be evaluated more thoroughly on neuroscience-relevant tasks?**
>
> **A7:**  We appreciate this constructive suggestion and agree that the core manuscript should remain squarely focused on modeling brain dynamics. To address this, we will: (1) Move the HAR experiment to a concise “**Method Validation**” section in the Supplement, so that the main text is entirely devoted to neuroscience applications. (2) Enrich our neuroscience evaluation by adding three EEG‑based benchmarks, such as Emotion recognition on the SEED [1], DEAP [2], and HCI [3] datasets, where our geometry‑aware SSM outperforms baseline methods (as below). We are committed to include the following EEG results in the final version.
>
> | **Mamba**     | **SEED**      | **DEAP**        | **HCI**       |
> |------------------|-----------------|-----------------|-----------------|
> | **Acc**         | 81.62 ± 0.48    | 88.98 ± 1.79    | 86.89 ± 4.56    |
> | **Pre**         | 79.76 ± 0.95    | 88.10 ± 1.83    | 75.51 ± 17.15    |
> | **F1**          | 76.31 ± 0.45    | 87.79 ± 2.07    | 69.78 ± 14.23    |
> | **Mamba**     | **SEED**      | **DEAP**        | **HCI**       |
> |------------------|-----------------|-----------------|-----------------|
> | **SPDNet**     | **SEED**      | **DEAP**        | **HCI**       |
> | **Acc**         | 85.32 ± 1.41    | 90.23 ± 2.40    | 93.16 ± 2.73    |
> | **Pre**         | 82.68 ± 1.88    | 88.69 ± 3.64    | 84.62 ± 6.21    |
> | **F1**          | 80.16 ± 1.92    | 88.48 ± 3.28    | 76.78 ± 2.54    |
> |------------------|-----------------|-----------------|-----------------|
> | **GeoDynamics**     | **SEED**      | **DEAP**        | **HCI**       |
> | **Acc**         | 92.51 ± 2.39    | 93.92 ± 1.46    | 95.23 ± 3.59    |
> | **Pre**         | 92.62 ± 2.36    | 93.94 ± 1.46    | 95.05 ± 3.74    |
> | **F1**          | 92.48 ± 2.39    | 93.91 ± 1.46    | 94.92 ± 4.05    |
>
> [1] Zheng, W. L., et al  (2015). Investigating critical frequency bands and channels for EEG-based emotion recognition with deep neural networks. IEEE TAMD.
>
> [2] Koelstra, S., et al. (2011). Deap: A database for emotion analysis; using physiological signals. IEEE TAC.
>
> [3] Soleymani, et al (2011). A multimodal database for affect recognition and implicit tagging. IEEE TAC.
>
> **Q8: Terminological inconsistency: Terms like "system state" vs. "hidden state" are used interchangeably, it would help to stick to one of them.**
>
> **A8:** Thank you for the suggestion. We have standardized the terminology throughout the manuscript by replacing all occurrences of both “system state” and “hidden state” with the single term “system state”. This change has been applied in Sections 2, 3, and all figure captions to ensure consistency.
>
> **Thanks again for your feedback.**

---

> > ### Comment · Reviewer_nqTq · 2025-08-07
> > **Thanks for your detailed rebuttal**
> >
> > Thank you for your thoughtful and detailed rebuttal.
> >
> > I appreciate the clarifications with regards to notation, which was one of my major concerns. I also appreciate the additional window size ablation study and addition of 3 EEG-based benchmarks.
> >
> > I strongly urge the authors to implement their suggested changes, and maintain a commitment to increasing the clarity of the manuscript and provide clean, consistent notation.
> >
> > In light of the clarifications given in the rebuttal, I will increase my score to (4).

---

> > > ### Author Response · Authors · 2025-08-07
> > >
> > > Dear, Reviewer,
> > >
> > > We sincerely appreciate your time and effort in acknowledging our response. We are **committed** to incorporating all of the reviewers’ suggestions into the final manuscript and are grateful for your feedback, as well as that of the other reviewers.
> > >
> > > Nice day!
> > >
> > > Warm regards,
> > >
> > > Authors

---

### Official Review · Reviewer_tBEG · 2025-07-03

**Clarity:** 3
**Significance:** 2
**Originality:** 3
**Rating:** 4
**Confidence:** 2

**Summary:**

The authors designed a geometric neural state-space neural networks (GeoDynamics) to model the functional connectivity (FC) among different brain areas. They benchmarked their model against fifteen other models (sequential and spatial types of model) and two datasets (HBC and HAR).

**Questions:**

L44 and Figure 1b legend: Does the "black solid box" indicate Euclidean space or Manifold space?

L251: So it is a 1 out 8 classification task?

Where is the Discussion of this paper? Section 4.3 is not the real Discussion.

**Ethical Concerns:**

["NO or VERY MINOR ethics concerns only"]

**Final Justification:**

The authors have answered my questions.

**Limitations:**

Yes (in supplementary)

**Paper Formatting Concerns:**

Some overlapped line numbers, such as 112+113, 177+178, 197+198, 233+234.

**Quality:**

3

**Strengths And Weaknesses:**

The benchmark against other models and coverage of datasets are comprehensive.

The presentation of Figures/Tables are clear.

My main concern is the improvement of current model over previous ones. In L325, the author said that "Our proposed GeoDynamics demonstrates a much better performance in these tasks". I believe they mean ADNI, OASIS and PPMI datasets. Are those improvement really big enoguh? For example, the Accuracy increase from 80.4 to 81.2 in ADNI (+1%), from 89.26 to 89.6 in OASIS (0.38%), and from 73.31 to 71.35 in PPMI (worse than previous models). What are the hyper-parameters of GeoDynamics?

Furthermore, are those values (Acc, Pre, and F1) shown in previous models copied from original papers or the authors reproduced previous models by themself?

---

> ### Author Rebuttal · Authors · 2025-07-30
>
> **We thank the reviewers for their constructive feedback and positive assessment of our manuscript. Below is a brief summary of our work, followed by our responses to the key concerns raised.**
>
> **Q1: Are those improvement really big enough? (the Accuracy increase from 80.4 to 81.2 in ADNI (+1%), from 89.26 to 89.6 in OASIS)**
>
> **A1:** Thank you for your thoughtful comment. While the observed accuracy gain may appear modest (e.g., 80.4% → 81.2% on ADNI), even small improvements can be meaningful in clinical or population-scale applications, especially when models are already operating near performance ceilings. To assess the robustness of this improvement, we conducted a paired t-test over the 10-fold cross-validation results. For ADNI, the improvement was statistically significant (p<0.05), suggesting the gain is unlikely due to random variation. For OASIS, however, the improvement is not signifiant at the level p<0.05. We also note important differences between the datasets. ADNI is more heterogeneous, spanning broader disease stages, acquisition protocols, and demographics—conditions under which modest gains are more likely to emerge and generalize. In contrast, OASIS is more homogeneous with higher baseline performance, making further improvement inherently more challenging.
>
> We have added this statistical analysis (Table 1 '*' denotes the significant improvement ($p<0.05$)) to the revised version and will include it in the final manuscript.
>
> **Q2: What are the hyper-parameters of GeoDynamics?**
>
> **A2:** The hyper-parameters are as follows: Learning rate: $5\times10^{-5}$, Weight decay=0, Batch size=16, Epochs=300, Hidden dim=number of brain region, network layer=2, window size=15. The full configuration is summarized in Table 5 of the original manuscript for reproducibility. We have conducted an ablation study in terms of difference window size (Appendix A.9 in the original manuscript). We will ensure this information remains clearly accessible in the final version.
>
> **Q3: Are those values (Acc, Pre, and F1) shown in previous models copied from original papers or the authors reproduced previous models by themself?**
>
> **A3:** Thank you for your question. All results reported for the baseline models—including accuracy, precision, and F1-score—were reproduced by us using publicly available implementations or reimplementations based on the descriptions in the original papers. To ensure a fair comparison, we applied the same 10-fold cross-validation splits, consistent preprocessing, and identical evaluation metrics across all models. This uniform setup ensures that performance differences reflect the intrinsic modeling capabilities rather than external confounding factors.
>
> We would like to clarify that the complete benchmarking code was already uploaded at the time of the initial submission to the anonymous GitHub repository (name "Comparison methods" folder) linked in the original manuscript. We respectfully encourage the reviewers to revisit the provided link for verification purposes. This inclusion complies with the rebuttal policy, as no new materials have been introduced. To further enhance transparency and reproducibility, we will make this information more prominent in the final version. Thank you so much.
>
> **Q4: L44 and Figure 1b legend: Does the "black solid box" indicate Euclidean space or Manifold space?**
>
> **A4**: In Fig.1b, the "black solid box" denotes the vanilla SSMs, which operate in Euclidean space. We agree that this distinction could be made more explicit, and we will revise the figure legend in the final version to clearly indicate that the “black solid box” corresponds to Euclidean space.
>
> **Q5: L251: So it is a 1 out 8 classification task?**
>
> **A5:** Yes, this is an 8-class classification task. We selected 1,081 subjects from the HCP-WM dataset, where the working memory paradigm included both 2-back and 0-back conditions across four stimulus categories: bodies, places, faces, and tools, along with fixation periods. The eight specific task events are: 2bk-body, 0bk-face, 2bk-tool, 0bk-body, 0bk-place, 2bk-face, 0bk-tool, and 2bk-place. A detailed description of the dataset and task design is provided in Appendix A.6 (Table 3, Page 18) of the original manuscript.
>
> **Q6: Where is the Discussion of this paper? Section 4.3 is not the real Discussion.**
>
> **A6:** Thank you for pointing this out. We have revised Section 4.3 to more accurately reflect its focus on **Model Validation**. The main discussion, including limitations, future directions, and broader implications, is provided in **Appendix A.10-A12 (Page 22 -23)** of the original manuscript. We will clarify this organization in the final version to improve readability and structure.
>
> **Q7: Format**
>
> **A7:**  Sorry about that. The overlapping of line numbers (e.g., 112+113, 177+178, 197+198, 233+234) results from our use of *\vspace* commands to condense vertical spacing. We will correct this and restore proper spacing in the final version of the manuscript.
>
> **Thank you for your time and consideration. We are committed to incorporating all suggestions and improvements into the final version of the manuscript.**

---

> > ### Comment · Reviewer_tBEG · 2025-08-05
> > **Thanks**
> >
> > Thank you for answering my questions and uploading your code during initial submission of the paper.
> >
> > I do not have further questions and have raised my score.

---

> > > ### Author Response · Authors · 2025-08-05
> > >
> > > We sincerely appreciate that.

---

### Decision · Program_Chairs · 2025-09-17

**Decision:**

Accept (poster)

**Comment:**

The authors propose a neural network model that integrates the temporal modeling of state-space models with geometric modeling, i.e. Riemannian manifold of SPD matrices, to learn a geometric representation of sequential data with inherent network structure. Their model is primarily targeted at modeling functional connectivity matrices in neural recordings (fMRI, EEG). GeoDynamics is demonstrated to have superior performance in multiple neuroscience datasets compared to a variety of baselines.

Four expert reviewers reviewed the paper.
The reviewers generally agree on the novelty of the proposed network model, the principled integration of Riemannian geometry in the approach, the significance of the neuroscience application studied, and the experimental results.
There were multiple concerns regarding clarity of the presentation, however, the reviewers agreed these were addressed by the detailed explanations during the rebuttal.
Additional experiments conducted during the rebuttal on EEG datasets were also received with positively.
After the rebuttal, the reviewers recommend acceptance.

The AC strongly encourages the authors to take into account all feedback from the reviewers regarding clarity and definitions, as discussed during the rebuttal period, to improve the final version of the manuscript.